# A Semi-Supervised Transfer Learning with Dynamic Associate Domain Adaptation for Human Activity Recognition Using WiFi Signals [note 1]

**DOI:** 10.3390/s21248475

**Published:** 2021-12-19

**Authors:** Yuh-Shyan Chen, Yu-Chi Chang, Chun-Yu Li

**Affiliations:** Department of Computer Science and Information Engineering, National Taipei University, No. 151, University Rd., San Shia District, New Taipei City 23741, Taiwan; s810976102@webmail.ntpu.edu.tw (Y.-C.C.); zx22960875@gmail.com (C.-Y.L.)

**Keywords:** human activity recognition, channel state information (CSI), semi-supervised learning, domain adaptation, attention

## Abstract

Human activity recognition without equipment plays a vital role in smart home applications, freeing humans from the shackles of wearable devices. In this paper, by using the channel state information (CSI) of the WiFi signal, semi-supervised transfer learning with dynamic associate domain adaptation is proposed for human activity recognition. In order to improve the CSI quality and denoising of CSI, we carried out missing packet filling, burst noise removal, background estimation, feature extraction, feature enhancement, and data augmentation in the data pre-processing stage. This paper considers the problem of environment-independent human activity recognition, also known as domain adaptation. The pre-trained model is trained from the source domain by collecting a complete labeled dataset of all of the CSI of human activity patterns. Then, the pre-trained model is transferred to the target environment through the semi-supervised transfer learning stage. Therefore, when humans move to different target domains, a partial labeled dataset of the target domain is required for fine-tuning. In this paper, we propose a dynamic associate domain adaptation called DADA. By modifying the existing associate domain adaptation algorithm, the target domain can provide a dynamic ratio of labeled dataset/unlabeled dataset, while the existing associate domain adaptation algorithm only allows target domains with the unlabeled dataset. The advantage of DADA is that it provides a dynamic strategy to eliminate different effects on different environments. In addition, we further designed an attention-based DenseNet model, or AD, as our training network, which is modified by an existing DenseNet by adding the attention function. The solution we proposed was simplified to DADA-AD throughout the paper. The experimental results show that for domain adaptation in different domains, the accuracy of human activity recognition of the DADA-AD scheme is 97.4%. It also shows that DADA-AD has advantages over existing semi-supervised learning schemes.

## 1. Introduction

Environmental sensors are widely deployed everywhere in our daily environment, with the environmental sensors including temperature, humidity, light, ultraviolet (UV) index, barometric pressure, noise, and acceleration with wireless communication functionality in an ultra-small footprint. With the environmental sensor data, it records our daily activities through human activity recognition (HAR) [1]. The research significance and practical value of HAR have attracted interest recently, so a large number of research results on HAR have been the focus of recent attention. The existing HAR systems usually use cameras, wearable devices, and sensors [2]. It is noticeable that vision-based and sensor-based fall detection had already been investigated in the literature. However, HAR using cameras had raised concerns about privacy violations and the limited coverage. In addition, wearable devices for HAR are not suitable for long-term monitoring because they are not easily accepted by the elderly, and their accuracy is often limited by environmental factors. The sensing device has a high reliability but cannot detect stationary events. However, all of the aforementioned methods require a large amount of hardware equipment with the limitation of the power lifetime to limit its universality. Consequently, it is very important and valuable to investigate the device-free HAR system.

As mobile smart devices and wireless networks affect all aspects of human production and life, wireless communication services (WCS) have gradually become an indispensable part of people’s lives. With the development of technology, WCS are no longer limited to providing communication services. Many existing studies have shown that WCS can be used in other fields, such as positioning, identification, and detection. Compared with other communication technologies, such as mmWave [3] and Bluetooth [4], the wavelength is shorter and the coverage is poor. With the proliferation of WiFi infrastructure, WiFi signals have become ubiquitous, especially in indoor scenarios. Due to the unguided nature of WiFi signal propagation, WiFi signals propagate freely in the atmosphere and may be reflected by walls or other objects. The multipath effect refers to the propagation of electromagnetic waves through different paths, with each component field arriving at the receiving end at different times and superimposing each other according to their respective phases to produce interference, distortion, or erroneous changes to the original signal.

Because human body interference changes the signal path, thereby affecting the transmission channel, a new human sensing and recognition technique can be initialized based on this feature. As a result, WiFi-based HAR has made great strides, and many efforts have been devoted to developing practical applications, for example, localization [5,6], motion prediction [7,8,9], fall prediction detection [10,11], etc. In a WiFi system, the received signal strength indicator (RSSI) only reflects the total amplitude of a multipath superposition, but CSI can show different subcarriers in orthogonal frequency division multiplexing. CSI can measure the frequency response of multiple subcarriers simultaneously from one data packet instead of superimposing all subcarriers. CSI illustrates the overall amplitude response, thereby more finely depicting the state of the channel. In addition, CSI extends the single-value RSSI to the frequency domain and adds phase information, which provides richer and fine-grained channel state information for wireless sensing in the frequency domain. However, there are still many challenges for using CSI. For instance, due to the different superposition of a multipath, the received signal of the same activity and the influence on the wireless channel are significantly different at different locations. In other words, activity recognition is relied on to find the location environment. In practical applications, the most urgent need is environment-independent recognition, which means that activities in any environment can be recognized. Assuming that smart devices for HAR can only be controlled in a fixed environment, there are huge obstacles in the promotion of technology applications.

Sigg and Orphomma et al. [7,8] used the change in RSSI signal to perform HAR. If the environment is complicated, the overall amplitude response superimposed by all of the subcarriers cannot achieve good sensing accuracy. Zhang et al. [9] used the change in CSI signal combined with a convolutional neural network-support vector machine (CNN-SVM) model for HAR, which achieves better results than that of only using RSSI and CSI signals. The Wifall system proposed by Wu et al. [10,11] uses the special diversity of CSI to detect human falls in indoor environments. The first version of Wifall [11] uses the subcarriers fusion method to reduce the amount of data. The modified version of Wifall [10] additionally considers the frequency as the weight index of the subcarriers fusion method. The FallSense [11] proposed by Huang performs the dynamic template matching (DTM) algorithm for Wifall [10,11]. These mentioned results rely on the complex classifiers such as SVM to produce the high computational complexity. All of these results point out that machine learning is a feasible solution for HAR.

CSI-based HAR can express more fine-grained information than RSSI-based HAR, but the impact of the environment reduces the recognition accuracy. Many studies are devoted to improve the recognition performance through signal processing technology. Gu et al. [12] proposed the Butterworth filter to filter out unreasonable high-frequency signals. Zhong et al. [13] used discrete wavelet transform (DWT) to denoise CSI raw data. The denoising method used for Li et al. [14] is the weighted moving average (WMA) method. Recently, deep learning (DL) has made utilized. Shi et al. [15] applied DL to extract features from CSI signals to improve the accuracy of action recognition. Yousef et al. [16] used a recurrent neural network (RNN) to extract features of the CSI signal, so that the representative features of the CSI signal were extracted. These above methods are affected by the CSI phase shift which is caused by the receiving time offset, called the carrier frequency offset (CFO) [17], by the distance between the transmitting antenna and the receiving antenna. It is noted that a slight error in the CSI phase may cause a significant reduction in the recognition rate. To estimate and compensate for the timing offset, some interesting results have been proposed. Wang et al. [18] evaluated the carrier frequency offset (CFO) between the transmitted signal and the received signal by considering the channel frequency response (CFR) power formula, and estimated the offset through the number of static and dynamic paths. After solving the time offset and signal noise, the uncertainty problem caused by environmental noise existed. In classification learning, the quality of the model has two indicators; the training sample and the new test sample strictly satisfy the independent and identical distribution and enough training samples to avoid underfitting. However, in practical applications, WiFi-based human activity recognition is difficult to meet these indicators. Zhenguo Shi et al. [19] proposed an environment-independent HAR using deep learning and enhanced CSI, called the HAR-MN-EF scheme. The HAR-MN-EF scheme is trained on a limited number of datasets from source environments and can directly recognized different activities in a new environment without retraining. Unfortunately, the HAR-MN-EF scheme does not provide better recognition accuracy, but the complexity of retraining is lower.

To improve the CSI quality and denoising of CSI of our work, we carried out burst noise removal, missing packet filling, background noise removal, and feature enhancement. In this paper, we further consider the problem of domain adaptation. The pre-trained model is trained from the source domain by a complete labeled dataset of CSI of human activity patterns. To significantly improve the recognition accuracy, efforts are made to slightly increase the retraining complexity by performing the fine-tune operation by adding the dynamic number of CSI with labeled data in the new environment. The pre-trained model is transferred to the target environment through the transfer learning stage. When humans move to different target domains, a partial labeled dataset of the target domain is allowed for the fine-tuning operation, depending on the target environment status.

Therefore, we specially design a dynamic associate domain adaptation, called DADA. By modifying the existing associate domain adaptation algorithm, the target domain can provide a dynamic ratio of the labeled dataset and the unlabeled dataset, while the existing associate domain adaptation algorithm only functions with the target domain without the labeled dataset. The advantage is that DADA provides a new strategy to dynamically eliminate different effects on different environments. The design of the dynamic capability of DADA is that it may allow the target domain to keep all of the labeled dataset or all of the unlabeled dataset, depending on the new target environment status.

In addition, we further designed an attention-based DenseNet model, or AD, as our training network, which modifies the existing DenseNet by adding the attention function. The solution we proposed was simplified to DADA-AD throughout the paper. The experimental results show that for domain adaptation in different domains, the accuracy of human activity recognition of the DADA-AD scheme is 97.4%. It also shows that DADA-AD has advantages over existing semi-supervised learning schemes.

As shown in Figure 1, the DADA-AD scheme is illustrated. Specifically, the main contributions of this work are as follows:An existing semi-learning learning work, called an associate domain adaption (ADA) scheme, is developed in [20], while the target domain is limited to be unlabeled dataset. The data of the source domain and the target domain are mapped in the same space through the similarity of relevance, under all data in the target domain that are unlabeled. To further provide a dynamically adjusted ratio of labeled and unlabeled datasets in the target domain, we modified the existing ADA algorithm to dynamic associative domain adaptation (DADA). We design a new semi-supervised transfer learning with dynamic associate domain adaption (DADA) capability for HAR. An improvement of this work is our proposed DADA scheme with the capability of a dynamically adjusted ratio, which is dynamically dependent on the target environment state. To improve the recognition accuracy, we may increase the dynamically adjusted ratio if the target domain encounters a new undesirable environment.The traditional ADA [20] has the limitation of data balance for the target domain, To overcome the problem of data imbalance, our proposed DADA also overcomes the problem of data imbalance. The data imbalance issue allows for the data imbalance to occur in our target environment. Our practical experimental results show that the average accuracy of DADA is 4.17% higher than that of ADA if there is data imbalance problem, but only 1.08% if there is no data imbalance problem.To increase the recognition accuracy, an attention-based DenseNet model (AD) is designed as our new training network. This modifies the existing DenseNet model and ECA-NeT (efficient channel attention-net) model. To reduce the data size, we adopted the bottleneck structure when ending denseblock and entering the next layer. It halves the data size and compresses the number of features. These operations may lose a lot of hidden information, so we adopt the ECA structure to retain important information. To avoid the loss of hidden information, we incorporate the ECA structure to strengthen the important channels of previous layers, and bring it to the next denseblock of DenseNet. Our experimental results show that the accuracy of AD as our training network is increased by 4.13%, compared to the existing HAR-MN-EF scheme [19].

The rest of the paper is organized as follows. Section 2 describes the research background, including the related works and motivation. Section 3 introduces the system model, problem expression, and basic ideas of this paper. Section 4 describes the proposed scheme. Section 5 discusses the performance results. Finally, Section 6 concludes this paper.

## 2. Background

This section firstly describes the related works in Section 2.1 and discusses the motivation of the research in Section 2.2.

### 2.1. Related Works

In this section, the recent research results on signal processing, including the denoising of a WiFi signal, and recognition technologies for WiFi-based HAR are introduced.

Noise removal is a key and important operation of HAR’s WiFi signal pre-processing. If redundant CSI is discarded, the same action characteristics can be retained. Gu et al. [12] proposed the Butterworth filter to attenuate unreasonable high-frequency signals using a fixed angular frequency and cut-off frequency. The bandwidth of the environmental instability signal is not fixed in a certain range, so this method may possibly simultaneously destroy the useful information, so cannot effectively perform the noise filtration. Zhong et al. [13] proposed discrete wavelet transform (DWT) to denoise CSI raw data, and use wavelet transform to denoise. When the signal and noise are decomposed by wavelet at different scales, the transfer characteristics shown are completely opposite. After wavelet decomposition, most of the wavelet coefficients with larger amplitude are useful signals, while the coefficients with smaller amplitudes are generally noise. The wavelet transform coefficients of useful signals can be considered to be greater than the wavelet transform coefficients of noise. When performing wavelet decomposition, parameter adjustment is a difficult problem. Although approximate estimation can solve this problem, it is still too complicated in terms of computational efficiency and parameter adjustment. It is observed that Li et al. [14] used the weighted moving average (WMA) method to solve this problem from a different perspective. The weighted moving average puts more weight on the most recent data, while exponentially attenuating the past data. This is done by multiplying the amplitude of each bar by a weighting factor, taking the calculated value as a fixed environmental noise, and subtracting it from the original CSI value to obtain the amplitude of the activity for calculation. Due to its unique calculation method, WMA reduces the amount of calculation and highlights the features more than DWT.

WiFi CSI signal has a strong time dependence, which also leads to too much data and too much information. How to extract useful features and use the simplified features for effective identification is an important issue. To solve this problem, Zou et al. [21] proposed DeepSense to design an autoencoder long-term recursive convolution network (AE-LRCN), which extracts the inherent time dependence through the long short-term memory (LSTM) module. Shi et al. [22] further proposed a CSI compensation and enhancement (CCE) method to compensate for the timing offset between the WiFi transmitter and receiver, enhance activity-related signals, and multiply the signal matrix to eliminate time information. Reduce the size of the signal input to the model, and the activity filter (AF) to distinguish similar activities has less training time and higher recognition accuracy. Li et al. [23] proposed a new solution by calculating the angle of arrival (ADoA) to eliminate position and background information. The phase difference of the same subcarrier in adjacent receiver antennas was calculated and measured from adjacent sample points. They extracted the principal components of ADoA to reduce the data dimension and simplify the training process.

When the environment changes, the background noise of the environment will change the characteristics, resulting in poor recognition efficiency. All of the above-mentioned research does not consider the environment change problem. Therefore, how to use as few samples as possible to achieve environment-independent sensing to achieve high-precision recognition is a crucial and quite critical issue. Shi et al. [19] proposed an environment-robust channel state information (CSI)-based HAR by leveraging the properties of a matching network (MatNet) and enhanced features, called HAR-MN-EF. This result achieves successful cross-environment HAR, and the MatNet is adopted to process features extracted by CSI-CE. MatNet allows to learn and extract inherent and transferable functions, thereby transferring knowledge in different environments. Unfortunately, although the knowledge of CSI information after feature extraction can be transferred, the required accuracy cannot be met only by directly transferring the features. Ding et al. [24] proposed a semi-supervised WiFi location-independent HAR, called WiLISensing. CNN architecture is used to identify activities in locations that do not require training or have few training samples through transfer learning methods by using a small number of sample transfer datasets to train the fully connected layer behind the network. This greatly reduces the need for training samples, but can improve recognition accuracy and convergence speed. Han et al. [25] proposed a semi-supervised, fine-grained, deep-adapted network gesture recognition scheme (DANGR), and GAN is used to expand the dataset. The key idea is to adopt the domain adaptation based on the multi-core maximum mean difference scheme. The mean embedding of the cross-domain abstract representation in the regenerated kernel Hilbert space is matched, and the deviation of the source domain and the target domain is compared, and the possible deviation of the distribution of CSI in various environments is reduced. Arshad et al. [26] also utilized transfer learning to develop a framework (TL-HAR) that can accurately detect a variety of human activities and use multiple-input multiple-output (MIMO) subcarrier variance to extract activity-based CSI.

### 2.2. Motivation

In order to be more practical, it is more important to support cross-environmental device-less or device-free CSI-based HAR research. The device-less HAR task based on WiFi CSI can provide people with high-quality, low-cost, and private human body monitoring services. Although the signal processing and identification methods in HAR have been already studied, there is room for efforts to solve the deviation of cross-environment CSI-based HAR with high recognition accuracy and low training cost. In practice, the relationship between CSI and the environment cannot be ignored because it may affect the prediction results.

Shi et al. [19] use matching network (MatNet) one-time learning technology to learn and extract inherent and transferable functions. MatNet uses one-shot learning technology (one shot learning) to efficiently transfer the environment, but the high-efficiency transfer makes the model’s recognition accuracy low; compared with DADA-AD using a small amount of data for fine-tuning, the recognition accuracy of DADA-AD transfer will be improved. Ding et al. [24] proposed WiLISensing, which uses supervised transfer learning to improve the efficiency of extracting transferable features, freezes the feature extraction layer, and learns from the label data brought into the target domain. They use label data to improved the recognition accuracy of MatNet, but in the process of learning transferable features, only the results of classification can be used to measure the result of learning classification features. Compared with DADA-AD by calculating similarity, DADA-AD can use a small number of the targets’ label data for transfer learning. Han et al. [25] used the results of classification to measure learning, and used the multicore maximum mean difference (MK-MMD) to measure the difference between domains, so as to provide a standard for the fusion of domain differences, thereby accelerating the efficiency of transfer learning. The need to calculate multiple kernel functions leads to poor model efficiency. DADA-AD can more accurately measure the difference between the two domains, and use the embedder to map, so that the time to calculate the similarity is linear. Therefore, this paper proposes the dynamic associate domain adaptation learning using attention-based DenseNet (DADA-AD) scheme to improve the generalization ability, maximize domain confusion, and minimize classification loss for source and target domains, as shown in Figure 2, combining the advantage of [19,24,25] to achieve the identification result that is not affected by the environment.

## 3. Preliminaries

This section describes the system model, problem formulation, and basic idea in Section 3.1, Section 3.2 and Section 3.3, respectively.

### 3.1. System Model

Figure 3 is our system architecture diagram of a semi-supervised transfer learning with dynamic associate domain adaptation for HAR using WiFi signals, which includes four modules; CSI data collection, raw data pre-treatment, data processing module, and activity recognition models. In this paper, five human activity patterns are considered; there are squat, sitting, stand, jump, and fall. The main design difficulty of HAR is multipath distortion caused by signal interference. We briefly describes these modules.

In CSI collection module, we collect CSI raw data through two computers equipped with Intel 5300 NIC as an interface by Intel IWL 5300 NIC tool [27]. MIMO communication technology of Modern COTS WiFi equipment is utilized, which can be equipped with multiple antennas for multiple inputs and multiple outputs, so we have $L=N_{r}\times N_{s}$ data streams. The Intel IWL 5300 NIC tool [27] can extract CSI raw data from *M* subcarriers of each pair of transceiver antennas, based on the IEEE 802.11n protocol, where M=30.

The raw data pre-treatment module is responsible for the handling with the data packet loss and sudden noise problem caused by the inconsistent transmission and reception power. The packet filter and noise filter operations are performed to recover from the abnormal amplitudes and data packet missing problems. In the data processing module, the activity features are captured and further perform the feature enhancement. All subcarriers of CSI data are used to calculate the correlation feature matrix, and the reduced data dimensionality of correlation feature matrix will be helpful to make the recognition model more robust.

In the activity recognition model we should consider the deviation of CSI distribution caused by different environments. This problem will be serious if data labeling is more difficult to obtain. The system structure is given in Figure 3, the CSI clean data is obtained from CSI raw data after executing the CSI collection model, raw data pre-treatment model, and data processing module. The CSI clean data is divided into source domain, denoted as *S*, and target domain, denoted as *T*. For the labeled data of the source domain, *S* is used to train the pre-training model. By transferring part of the knowledge learned from the pre-training model, the labeled data and unlabeled data of target domains are associated to bridge the difference between the two different environmental data.

### 3.2. Problem Formulation

We follow the similar notations defined in [19], let h(t) be magnitude of CSI vector at *t*-th packet, which can be given by
(1)$$\begin{array}{c} h\left(t\right)={\left[h_{1,1}\left(t\right),\ldots ,h_{1,m}\left(t\right),\ldots ,h_{l,m}\left(t\right),\ldots ,h_{L,M}\left(t\right)\right]}^{T} \end{array}$$
where $h_{1,m}\left(t\right)$ is the CSI information in the l-th wireless link for *m*-th subcarrier of *t*-th packet, *M* is the total number of subcarriers in each wireless link; *L* is the total number of wireless links, where $L=N_{r}\times N_{s}$, $N_{r}$ and $N_{s}$ are the number of transmitter and receiver antennas. T denotes the transpose operation. In this paper, the Intel IWL 5300 NIC tool [27] is used to extract CSI information from *M* subcarriers of each pair of transceiver antennas, based on the IEEE 802.11n protocol, where M=30. After collecting CSI vectors from *K* packets, the CSI matrix *H* can be expressed as:(2)$$\begin{array}{c} H=[h(1),\ldots ,h(t),\ldots ,h(K)]. \end{array}$$

In order to improve the CSI quality and denoising of CSI, a new CSI matrix $\tilde{H}$ is obtained by carried out burst noise removal, missing packet filling, background noise removal, and feature enhancement operations on CSI matrix H=[h(1),…,h(t),…,h(K)] in the data pre-processing stage, which will be described in detail in Section 4.1. In this work, $D_{s}$ and $D_{t}$ represent the source domain and the target domain, ${\tilde{H}}^{s}$ and ${\tilde{H}}^{t}$ are further denoted as the CSI matrix from the source domain and the target domain, respectively. We consider a source domain, $D_{s}={\left\{{\tilde{H}}_{i}^{s},y_{i}^{s}\right\}}_{\left(i=1,\ldots ,n_{s}\right)}$, where ${\tilde{H}}_{i}^{s}$ is the *i*-th collected CSI matrix ${\tilde{H}}_{i}^{s}$ from the source environment, and $y_{i}^{s}$ is the corresponding label of ${\tilde{H}}_{i}^{s}$. The target domain, $D_{t}={\left\{{\tilde{H}}_{i}^{t},y_{i}^{t}\right\}}_{\left(i=1,\ldots ,n\right)}\cup {\left\{{\tilde{H}}_{i}^{t}\right\}}_{\left(i=n+1,\ldots ,n_{t}\right)}$, $n^{s}$ and $n^{t}$ represent the total number of $D_{s}$ and $D_{t}$ data, respectively, where ${\tilde{H}}_{i}^{t}$ is the *i*-th collected CSI matrix $\tilde{H}t$ from the target environment. It is observed that CSI matrix ${\tilde{H}}_{i}^{t}$ of the target environment has target label $y_{i}^{t}$ where 1≤i≤n. However, there are no target labels for all ${\tilde{H}}_{i}^{t}$ where $n+1\leq i\leq n_{t}$. That is, the target labels ${\left\{y_{i}^{t}\right\}}_{\left(i=n+1,\ldots n_{t}\right)}$ are not available for training.

It is noted that the source domain, $D_{s}={\left\{{\tilde{H}}_{i}^{s},y_{i}^{s}\right\}}_{(i=1,\ldots ,n_{s})}$, and the target domain, $D_{t}={\left\{{\tilde{H}}_{i}^{t},y_{i}^{t}\right\}}_{\left(i=1,\ldots ,n\right)}\cup {\left\{{\tilde{H}}_{i}^{t}\right\}}_{\left(i=n+1,\ldots ,n_{t}\right)}$, $n^{s}$ are associated with the same labeled space due to the same activity patterns that are kept in the source domain and any other target domains.

Efforts will be made in this work to re-design the association between the source domain and the target domain through the statistical similarity measures. Basically, the source and the target domains have similar features, while it may reduce the prediction error from the source domain to the target domain. The source and target domains come from different distributions, and these influences are opposed to each other. This work is divided into two parts; the pre-training stage and the fine-tuning stage.

In the first part, we formalized the objective function $L_{p}$ in the pre-training stage as follows:(3)$$\begin{array}{c} arg\;\times \;min\;L_{p}\\ L_{p}=\frac{1}{n_{s}}\sum_{i=1}^{n_{s}}H(y_{i}^{s},p_{i}^{s})=\frac{1}{n_{s}\times m}\sum_{i=1}^{n_{s}}\sum_{j=1}^{m}y_{i,j}^{s}\times log\left(p_{i,j}^{s}\right)\\ subject\;to\left\{\begin{array}{ccc} y_{i,j}^{s}\; & ,\;y_{i,j}^{l}\;,\; & \forall \;i,j\\ {0\leq p}_{i,j}^{s} & ,p_{i,j}^{l}\leq 1\;,\; & \forall \;i,j \end{array}\right. \end{array}$$
where $Q(y_{i}^{s},p_{i}^{s})$ is the cross-entropy used for the classification problem in the source domain, $y_{i}^{s}$ is denoted as the *i*-th data belonging to the real category in the source domain, and $p_{i}^{s}$ is denoted as the predicted probability in the source domain, $n_{s}$ represent the number of training data of the source domain, *m* is defined as the number of classification categories, $y_{i,j}^{s}$ is denoted as the *i*-th data belonging to the real category of the *j*-th category in the source domain, and $p_{i,j}^{s}$ is denoted as the predicted probability; the purpose of $L_{p}$ is to minimize the classification error.

In the second part, we formalized the objective function $L_{f}$ in the fine-tuning stage as follows:(4)$$\begin{array}{c} arg\;\times \;min\;L_{f}\\ L_{f}=\left(1-\lambda \right)L_{c}+{\lambda \times L}_{sim}\\ subject\;to\;\;0\leq \lambda \leq 1 \end{array}$$
where $L_{f}$ is the combined objective function of both considering the $L_{c}$ and $L_{sim}$, where $L_{c}$ is the objective function of classification, $L_{sim}$ is the objective function of the similarity problem, $L_{f}$ is the weight-sum of $L_{c}$ and $L_{sim}$, as the total objective function, where λ is the hyper-parameter of the hybrid objective function. The objective function $L_{sim}$ is used to measure the difference between two different distributions, which is expressed as Wasserstein
distance below,
(5)$$\begin{array}{c} L_{sim}=\sqrt{\underset{P_{st}}{min}E_{P_{st}}{∥\phi \left(D_{s}\right)-\phi \left(D_{t}\right)∥}_{2}^{2}}. \end{array}$$

The distribution $P_{s}$ represents the distribution of $D_{s}$, the distribution $P_{t}$ represents the distribution of $D_{t}$,and $P_{st}$ is the joint distribution of $P_{t}$ and $P_{t}$, and $P_{s}\neq P_{t}$, ϕ represents a mapping function, which maps data of different distributions to the same space. $L_{sim}$ represents is the joint distribution $P_{st}$. Find out the minimum expected value $E_{P_{st}}$ by mapping $D_{s}$ and $D_{t}$ in the same space through ϕ.

$L_{c}$ is the objective function used for classification problems in the fine-tuning stage and is given as follows:(6)$$\begin{array}{c} L_{c}=max\;\left[\frac{1}{N_{s}}\sum_{i=1}^{N_{s}}H(y_{i}^{s},p_{i}^{s}),\frac{1}{N_{l}}\sum_{i=1}^{N_{l}}H(y_{i}^{l},p_{i}^{l})\right] \end{array}$$
(7)$$\begin{array}{c} H\left(y_{i}^{s},p_{i}^{s}\right)=-\frac{1}{m}\sum_{j=1}^{m}y_{i,j}^{s}\times log{(p}_{i,j}^{,}) \end{array}$$
(8)$$\begin{array}{c} H\left(y_{i}^{l},p_{i}^{l}\right)=-\frac{1}{m}\sum_{j=1}^{m}y_{i,j}^{l}\times log{(p}_{i,j}^{l}) \end{array}$$
$$subject\;to\left\{\begin{array}{ccc} y_{i,j}^{s}\; & ,\;y_{i,j}^{l}\;,\; & \forall \;i,j\\ {0\leq p}_{i,j}^{s} & ,p_{i,j}^{l}\leq 1\;,\; & \forall \;i,j \end{array}\right.$$
where $Q(y_{i}^{s},p_{i}^{s})$ and $Q(y_{i}^{t},p_{i}^{t})$ are cross-entropy function of source domain and target domain for classification, where $Q(y_{i}^{s},p_{i}^{s})$ is the cross-entropy used for classification problem in the source domain, $y_{i}^{s}$ is denoted as the i-th data belonging to the real category in the source domain, and $p_{i}^{s}$ is denoted as the predicted probability in the source domain, $Q(y_{i}^{t},p_{i}^{t})$ is the cross-entropy used for rhe classification problem in the target domain, $y_{i}^{t}$ is denoted as the *i*-th data belonging to the real category in the target domain, and $p_{i}^{t}$ is denoted as the predicted probability in the target domain, and $n_{s}$ and $n_{t}$ represent the number of training data of the source and target domains. m is defined as the number of classification categories, $y_{i,j}^{s}$ and $y_{i,j}^{t}$ are respectively denoted as the *i*-th data in $D_{s}$ and $D_{t}$, the data belonging to the real category of the *j*-th category. $p_{i,j}^{s}$ and $p_{i,j}^{t}$ are expressed as the predicted probabilities that belong to the predict category of the *j*-th category of $D_{s}$ and $D_{t}$, respectively. Since the purpose of the model is to bridge the two domains, $D_{s}$ and $D_{t}$ can be regarded as the same distribution, and $L_{c}$ is to reflect the worst state of the split and provide the model for adjustment, and the purpose of $L_{c}$ is to minimize the classification error.

### 3.3. Basic Idea

The basic idea is to use the domain adaptation technique to transfer the common features. Figure 4 shows that the process is divided into two parts. The first part is the pre-training part. The data enhancement is adopted for the data in the source domain to produce the pre-training model. The second part is to transfer the knowledge in the pre-training to find out the features shared by the source domain and the target domain. With t-SNE embedding technique, it is found that the shared features exists in the shallower model. The knowledge of the shallower model is transferred so that the characteristics of the two domains can be fitted. Figure 5 and Figure 6 show the design differences. Figure 5 shows [25] using the MK-MMD technique for transfer learning; multiple kernels are used to project the features on the kernel Hilbert space, and the distance between the average embeddings of the two probability distributions is calculated. This distance is calculated by the kernel tricks.

Figure 6a shows the semi-supervised associative domain adaptation (ADA) proposed by [20], which mainly embeds features in the same space and uses associative algorithms to map the features of the source domain to the target domain, is being mapped back from the target domain, but it is impossible to measure whether the features of the mapped target domain are evenly mapped, so the training model must ensure that the target domain is evenly distributed. Figure 6b shows the dynamic associative domain adaptation (DADA) proposed in this paper.

## 4. A Semi-Supervised Transfer Learning with Dynamic Associate Domain Adaptation for HAR

In this section, we propose a HAR algorithm based on semi-supervised dynamic associate domain adaptation learning in WiFi networks to predict unlabeled activity recognition with the cross-domain data. The flowchart of the proposed algorithm is given in Figure 7. The algorithm is divided into four phases.

(1)**Data collection and processing phase**: This phase aims to collect CSI data by keeping the environment reinforcement, and avoiding the hardware defects. The main work of this phase is to discard redundant information, retain the characteristics of enhanced activity, and reduce irrelevant information.(2)**Pre-training phase**: This phase is to build an attention-based DenseNet (AD), as our training network. In AD, DenseNet is adopted as backbone network and further add the ECA structure to retain the important training information. The activity classification is pre-trained in this phase through feature reuse and attention mechanism for the transfer training.(3)**Dynamic associate domain adaptation phase**: This phase aims to project the features of two different domains into the same space through DNN embedding by using a dynamic associate domain adaptation algorithm (DADA). The dynamic associate domain adaptation is to improve previous work, associate domain adaptation, to further consider the data imbalance problem. In addition, our dynamic associate domain adaptation can dynamically adjust the ratio of labeled dataset/unlabeled dataset.(4)**Associate knowledge fine-tuning phase**: In this phase, the HAR through the image has the characteristics of domain invariance, the weights of the shallow layers of the source domain learned previously are unchanged and frozen as a common feature, and the knowledge of the deep layers is transferred to new target domain for fine-tuning.

### 4.1. Data Collection and Processing Phase

The main task of Phase 1 is to collect and process WiFi CSI data. Modern COTS WiFi equipment is with MIMO communication technology, which enables it to be equipped with multiple antennas for multiple inputs and multiple outputs. About the WiFi CSI data, this paper uses the image recognition method for human activity recognition; the phase information of CSI is not useful of our processing work, and the phase offset of CSI can be ignored. Amplitude intensity of CSI will be completed utilized in this work. As defined in Section 3.2, after collecting CSI vectors from *K* packets, the CSI matrix *H* can be expressed as, H=[h(1),…,h(i),…,h(K)], for 1≤i≤K, and $h\left(i\right)={\left[h_{1,1}\left(i\right),\ldots ,h_{1,k}\left(i\right),\ldots ,h_{j,k}\left(i\right),\ldots ,h_{L,M}\left(i\right)\right]}^{T}$, where 1≤j≤L and 1≤k≤M. Consequently, CSI matrix *H* is performed in six steps, including missing packet filling, burst noise removal, background estimation, feature extraction, feature enhancement, and data augmentation operations, before the next phase. The details of the CSI data collection and data pre-processing phase are described as follows:
**S1.** 
**Missing packet filling:** To solve the packet loss problem, a timer is set in RX, and the timer starts after the packet is received. To maintain the continuity of the signal, the linear interpolation is used to repair the lost packets. Assuming $h_{j,k}\left(i\right)$ is a lost packet of H=[h(1),…,h(i),…,h(K)], for 1≤i≤K, and $h\left(i\right)=[h_{1,1}\left(i\right),\ldots ,h_{1,k}\left(i\right),\ldots ,h_{j,k}\left(i\right),\ldots ,$${h_{L,M}\left(i\right)]}^{T}$, where 1≤j≤L and 1≤k≤M. The lost packet $h_{j,k}\left(i\right)$ can be repaired by a simple linear interpolation function as:
(9)$$\begin{array}{c} h_{j,k}\left(i\right)=\left(i-p\right)\frac{h_{j,k}\left(n\right)-h_{j,k}\left(p\right)}{n-p}+h_{j,k}\left(p\right) \end{array}$$
where $h_{j,k}\left(p\right)$ and $h_{j,k}\left(n\right)$ are represented as the previous packet and the next packet of $h_{j,k}\left(i\right)$, respectively. The output matrix $H^{pf}=\left[h^{pf}\left(1\right),\ldots ,h^{pf}\left(i\right),\ldots ,h^{pf}\left(K\right)\right]$,where 1≤i≤K, is obtained, where $H^{pf}=liner\_\;interpolation\left(H\right)$.**S2.** 
**Burst noise removal:** To perform the burst noise removal operation on $H^{pf}$ matrix due to the sudden noise caused by the environment and hardware equipment, we adopt the wavelet transform denoising [28] algorithm to $H^{pf}$ matrix to obtain $H^{nr}$ matrix as follow:
(10)$$\begin{array}{c} H^{nr}=DWT\left(o,p,H^{pf}\right)=\int_{-\infty }^{\infty }2^{-\frac{o}{2}}\psi \left(2^{-o}i-p\right)h^{pf}\left(i\right)\;di \end{array}$$A six-level discrete wavelet transform is used to decompose, and symlet is used as the wavelet base, and the denoised CSI packet sequence will be reconstructed through inverse transform. The output matrix $H^{nr}=\left[h^{nr}\left(1\right),\ldots ,h^{nr}\left(i\right),\ldots ,h^{nr}\left(K\right)\right]$ is obtained for 1≤i≤K.**S3.** 
**Background estimation:** There exists some useless background information of human activities in matrix $H^{nr}=\left[h^{nr}\left(1\right),\ldots ,h^{nr}\left(i\right),\ldots ,h^{nr}\left(K\right)\right]$ for 1≤i≤K, which are not related to human activities. These useless feature may reduce the quality of the trained model. We let $h^{nr}\left(i\right)$ be represented as $h^{nr}\left(i\right)=h^{be}\left(i\right)+h^{fe}\left(i\right)$ for 1≤i≤K. The useless background information is represented by $h^{be}\left(i\right)$, which is also denoted as the static CSI vector at time *i*, $h^{f}e\left(i\right)$ represents as the dynamic CSI vectors, which is represented as the useful features of the human motion at time *i*, where $h^{fe}\left(i\right)$ is obtained by a number of activity-related features. The main work is to estimate the dynamic CSI vector $h^{fe}\left(i\right)$, being generated by the human activities, so $h^{be}\left(i\right)$ is initially obtained for 1≤i≤K by adopting the exponentially weighted moving average (EWMA) algorithm [29] as follows:
(11)$$\begin{array}{c} h^{be}\left(i\right)=\lambda h^{nr}+\left(\lambda -1\right)h^{be}\left(i-1\right) \end{array}$$
where 1≤i≤K, λ is the forgetting factor, where 0≤λ≤1. Each new estimated point is recursively calculated from the previous observations and attenuated by a forgetting factor. Consequently, the static CSI matrix $H^{be}=\left[h^{be}\left(1\right),\ldots ,h^{be}\left(i\right),\ldots ,h^{be}\left(K\right)\right]$, where 1≤i≤K.**S4.** 
If static CSI matrix $H^{be}=\left[h^{be}\left(1\right),\ldots ,h^{be}\left(i\right),\ldots ,h^{be}\left(K\right)\right]$, where 1≤i≤K, is finding, so the dynamic CSI matrix $H^{fe}$ is obtained by $H^{fe}=H^{nr}-H^{be}$, where $H^{fe}=\left[h^{fe}\left(1\right),\ldots ,h^{fe}\left(i\right),\ldots ,h^{fe}\left(K\right)\right]$.**S5.** 
**Feature enhancement:** The matrix size of the dynamic CSI matrix $H^{fe}$ is L×M and a width of 1 kHz ×3 s. We adopt the similar feature enhancement algorithm [19] to obtain the correlation matrix $\tilde{H}$, where $\tilde{H}={H^{fe}\times H^{fe}}^{T}$, to enhance the correlation between the signals on the subcarriers, which is more important than the time dimension data. The correlation matrix between the signals on all subcarriers *M* eliminates the time dimension information, leaving the characteristics of the correlation between the subcarriers. The matrix size of $\tilde{H}$ is smaller than the original size of $H^{fe}$, which is also reduced in the complexity of the trained model.
(12)$$\begin{array}{c} \tilde{H}={H^{fe}\times H^{fe}}^{T}=\left[\begin{array}{ccccc} h^{fe}\left(1\right)\times h^{fe}\left(1\right) & \cdots & h^{fe}\left(1\right)\times h^{fe}\left(i\right) & \cdots & h^{fe}\left(1\right)\times h^{fe}\left(K\right)\\ \vdots & \ddots & \vdots & \ddots & \vdots \\ h^{fe}\left(i\right)\times h^{fe}\left(1\right) & \cdots & h^{fe}\left(i\right)\times h^{fe}\left(i\right) & \cdots & h^{fe}\left(i\right)\times h^{fe}\left(K\right)\\ \vdots & \ddots & \vdots & \ddots & \vdots \\ h^{fe}\left(K\right)\times h^{fe}\left(1\right) & \cdots & h^{fe}\left(K\right)\times h^{fe}\left(i\right) & \cdots & h^{fe}\left(K\right)\times h^{fe}\left(K\right) \end{array}\right]\; \end{array}$$**S6.** 
**Data augmentation:** To enhance the robustness of model training, the data augmentation technique is used to enlarge the training dataset to generate more training data. In this work, the correlation matrix $\tilde{H}$ will be augmented by adopting the spin, mask, and zoom methods.

In this work, we use one transmitting antenna ($N_{r}=1$) and three receiving antennas ($N_{s}=3$) for five kinds of activity recognition; standing, sitting, squatting, jumping, and falling. To more easily highlight the difference of each stages, a 2D diagram is converted into a 3D diagram of all the CSI matrix in Figure 8, and the color of the 2D diagram is restored to the amplitude. The sampling rate is 1 kHz and sampling is 3 s, the CSI matrix size is 90×3000, where the Intel IWL 5300 NIC tool extracts CSI raw data from M=30 subcarriers of three pairs of transceiver antennas ($N_{r}=1$ and $N_{s}=3$) in our experimental environment. By using linear interpolation to repair the missing data packets, and also using the wavelet transform to remove noise, Figure 8a is an example of the package filling operation of matrix $H^{pf}$ for the fall activity pattern. The burst noise removal operation is performed by using the EWMA algorithm, and the output matrix $H^{nr}$ is obtained. Figure 8b is an example of denoising CSI matrix $H^{nr}$ of the burst noise removal operation. Figure 8c is an example of static CSI matrix $H^{be}$ of the background estimation operation. Figure 8d is an example of dynamic CSI matrix $H^{fe}$ of the feature enhancement operation. Figure 8e is an example of the correlation matrix $\tilde{H}$ of the feature enhancement operation. We observe that Figure 8 shows the 3D diagram of the data matrix, where the *x*-axis is subcarriers, *y*-axis is time, and *z*-axis is amplitude. We also provide a 2D diagram of the each data matrix in Figure 9, value of (x,y) of the 2D diagram is its amplitude, where the *x*-axis is subcarriers and the *y*-axis is time.

### 4.2. Pre-Training Phase

The 2D correlation matrix,
$\tilde{H}$, between the signals is obtained to reduce the data complexity of the training. The correlation matrix $\tilde{H}$ will be used to pre-training the knowledge for the source domain. Note that the pre-training dataset is the source domain, $D_{s}={\left\{{\tilde{H}}_{i}^{s},y_{i}^{s}\right\}}_{\left(i=1,\ldots ,n_{s}\right)}$, where ${\tilde{H}}_{i}^{s}$ is the *i*-th collected CSI matrix ${\tilde{H}}_{i}^{s}$ from the source environment, and $y_{i}^{s}$ is the corresponding label of ${\tilde{H}}_{i}^{s}$. To increase the recognition accuracy, an attention-based DenseNet (AD) model is designed as our new training network. This modified the existing DenseNet model and ECA-NeT (efficient channel attention-net) model. To reduce the data size, we adopted the bottleneck structure, when ending denseblock and entering the next layer. It halves the data size and compresses the number of features. These operations may lose a lot of hidden information, so we adopt the ECA structure to retain important information. To avoid the loss of hidden information, we incorporate ECA structure to strengthen the important channels of previous layers, and bring it to the next denseblock of DenseNet. The algorithm is given in Algorithm 1 and the corresponding operations of Algorithm 1 are also given below.

**S1.** 
The basic training network of this work adopted a deep DenseNet model [30]. A deep DenseNet is constructed by a number of denseblocks. The dense connection [30] is also utilized, where each layer is repeatedly connected with all of the previous layers in the channel dimension. Note that denseblock directly connects feature maps from different layers. In denseblock, the output of all of the previous layers is connected as input, $z_{o}^{D}$, for the next layer, and can be expressed as:
(13)$$\begin{array}{c} z_{o}^{D}=\sigma \left(D_{0},\ldots ,D_{L-1}\right) \end{array}$$
where $\sigma_{D}$ represents a non-linear transformation function, and $D_{L-1}$ represents the output of the L−1 layer in the denseblock. Each convolutional layer produces different feature maps. All of the feature maps obtained each time are called a channel. Assuming that each layer in the denseblock uses *k* convolution kernels, we set the growth rate to be *k*. Let the channel number of the feature map in the input layer be $c_{0}$, and the last output channel number is
(14)$$\begin{array}{c} C^{D}=c_{0}+k^{D}\left(L-1\right) \end{array}$$
where denseblock utilizes the bottleneck architecture to reduce the calculation cost. Specially, it is noted that each layer produces *k* output feature maps, and is the same as that of the growth rate (=*k*) and convolution kernels (=*k*).**S2.** 
To increase the recognition accuracy, an attention-based DenseNet (AD) model is designed as our new training network. This is the modified existing DenseNet model and ECA-NeT (efficient channel attention-net) model. This modified work is done as follows. The feature is extracted through the dense connection mechanism. For connecting two adjacent denseblock, they are connected by using ECAT architecture [31], while ECAT is a connection layer based on the channel attention mechanism ECA architecture [31]. Because the denseblock input channel number is determined by the number of channels of the denseblock in the upper layer, if the channel dimension is not reduced through the connection layer, this leads to too many parameters and inefficient calculation. We expect to add the channel attention mechanism to strengthen the correlation between feature channels to improve training accuracy. The feature map with the same size is maintained without destroying the features. The number of channels is reduced and the size of the feature map is halved. Suppose there are $C^{D}$ input channels, the number of output channels, $C^{T}$, of a denseblock is expressed as:
(15)$$\begin{array}{c} C^{T}=\theta C^{D} \end{array}$$
where 0<θ≤1, θ is the compression factor. The output feature, $z_{o}^{T}$, by the connection layer is expressed as:
(16)$$\begin{array}{c} z_{o}^{T}=\sigma \left(z_{o}^{D}\right) \end{array}$$
where $\sigma_{T}$ represents the non-linear transformation, which is repeatedly used in the transition layer [30], by adding the ECA network as a substructure, it is embedded in the connection layer to learn feature weights to achieve better training results.Through the global average pooling, we designed it to be flattened into $1\times 1\times C^{T}$. Through the convolution of 1×1×cs, the mutual relationship between each channel and its cs neighboring channels is constructed, where cs is related to the channel dimension $C^{T}$. The larger the number of channels is, the stronger the relationship to adjacent channels will be. The relationship can be expressed as $C^{T}=\psi \left(cs\right)$, where ψ is the approximate exponential mapping function, and is expressed as $\psi \left(cs\right)=2^{\left(\gamma \;\times \;cs-\omega \right)}$, given the channel dimension $C^{T}$, the adaptation channel size cs [31], i.e., the number of neighboring channels, is expressed as:
(17)$$\begin{array}{c} cs={\left|\frac{{log}_{2}\left(C^{T}\right)+\omega }{\gamma }\right|}_{odd} \end{array}$$
where ω and γ is set as 1 and 2. With ECAT, the channel and feature size is adjusted and the channel dimension is also reduced, and the important channel weight can be increased through the channel attention mechanism. The weight is expressed as:
(18)$$\begin{array}{c} w_{o}=\sigma_{ECA}^{cs}\left(z_{o}^{T}\right) \end{array}$$
where $\sigma_{ECA}^{cs}$ is an adaptation non-linear transformation which is composed of global average pooling and 1×1×cs convolution. Consequently, the output weighted feature $z_{o}^{ECAT}$ is
(19)$$\begin{array}{c} z_{o}^{ECAT}=z_{o}^{T}\;\times \;w_{o} \end{array}$$
by multiplying weight $w_{o}$ and output $z_{o}^{T}$ of the connection layer. After repeating the denseblock structures with the ECAT mechanism twice, the maxpooling operation MP is applied to the feature to extract the maximum value.
(20)$$\begin{array}{c} m_{o}=MP\left(z_{o}^{p}\right) \end{array}$$
where $z_{o}^{p}$ is the final output before reaching the flattening layer, where $m_{o}$ is the flattened feature. Finally, $m_{o}$ pass through an *f*-layer fully connected layer to obtain the final feature, denoted as d and used for the final activity prediction through a fully connected layer by an activation function of softmax.
(21)$$\begin{array}{c} p_{i}^{s}=W_{o}\times d+b_{o} \end{array}$$
where $p_{i}^{s}$ represents the predicted value of ${\tilde{H}}_{i}^{s}$, where $W_{o}$ and $b_{o}$ are trainable parameters.**S3.** 
After obtaining the final activity prediction, the loss function is calculated through cross-entropy based on the actual label $y_{i}^{s}$ and the activity prediction $p_{i}^{s}$ obtained in the previous phase $L_{p}$.
(22)$$\begin{array}{c} arg\;\times \;min\;L_{p}\\ L_{p}=\frac{1}{n_{s}}\sum_{i=1}^{n_{s}}H(y_{i}^{s},p_{i}^{s})=\frac{1}{n_{s}\times m}\sum_{i=1}^{n_{s}}\sum_{j=1}^{m}y_{i,j}^{s}\times log\left(p_{i,j}^{s}\right)\\ subject\;to\left\{\begin{array}{ccc} y_{i,j}^{s}\; & ,\;y_{i,j}^{l}\;,\; & \forall \;i,j\\ {0\leq p}_{i,j}^{s} & ,p_{i,j}^{l}\leq 1\;,\; & \forall \;i,j \end{array}\right. \end{array}$$
where $n_{s}$ is the number of training data in the source domain, *m* is the number of classification categories, $y_{i,j}^{s}$ is expressed as the true category of the *i*-th data in $D_{s}$ that belongs to the *j*-th category, $p_{i,j}^{s}$ is expressed as the prediction of the *i*-th data in $D_{s}$ belonging to the *j*-th category.

The attention-based DenseNet (AD) model is shown in Figure 10a. Figure 10b shows that the AD model uses the bottleneck architecture to reduce the calculation cost, while maintaining the same size of feature map without destroying the features, and reduce the channel number. For instance, we let
$\sigma_{D}$ use BN + ReLU + 1 × 1 convolution + BN + ReLU + 3 × 3 convolution, in addition to 1 × 1 convolution before 3 × 3 convolution. Figure 10c is the detailed ECAT structure, while the channel number is reduced and the size of the feature map is halved. Let
$\sigma_{T}$ be a non-linear transformation which is composed of 1 × 1 ×
$C^{T}$ convolution and 2 × 2 pooling.

The time complexity of the model structure is linear, and the time complexity of calculating the cross-entropy is $n_{s}\;\times \;m$, where
$n_{s}$ is the number of source data, and *m* is the number of categories, the number of categories is a fixed constant in this work, so the overall time complexity in Algorithm 1 is
$O(n_{s})$.

**Algorithm 1:** The pre-training phase.

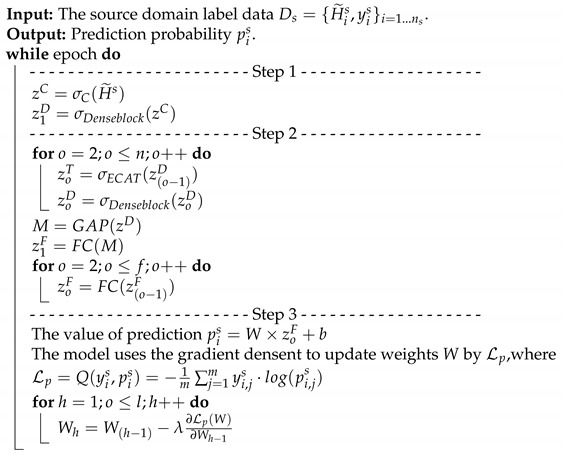



### 4.3. Dynamic Associate Domain Adaptation Phase

A novel semi-learning learning work, called an associate domain adaption (ADA) scheme, is developed in [20]. The task of inferring class labels for an unlabeled target domain of an ADA scheme based on the statistical properties of a labeled source domain. The data of the source domain and the target domain are mapped in the same space through the similarity of relevance. We modified the existing ADA algorithm to a dynamic associative domain adaptation (DADA). One improvement of this work is our proposed DADA scheme can dynamically adjust the ratio of a labeled dataset and an unlabeled dataset of the target domain, which is dynamically dependent on the target environment status. It is noted that all of the target domain is unlabeled on the ADA scheme. In order to improve accuracy, if the target domain encounters a new bad environment, we may increase the ratio of the labeled dataset and the unlabeled dataset of the target domain. The traditional ADA [20] has the limitation of a data balance for the target domain, To overcome the problem of the data imbalance, DADA is proposed. The data imbalance issue allows for the data imbalance to occur in our target environment.

Recall the notations, the source domain $D_{s}={\left\{{\tilde{H}}_{i}^{s},y_{i}^{s}\right\}}_{\left(i=1,\ldots ,n_{s}\right)}$, where ${\tilde{H}}_{i}^{s}$ is the *i*-th collected CSI matrix ${\tilde{H}}_{i}^{s}$ from the source domain, and $y_{i}^{s}$ is the corresponding label of ${\tilde{H}}_{i}^{s}$. The target domain $D_{t}={\left\{{\tilde{H}}_{i}^{t},y_{i}^{t}\right\}}_{\left(i=1,\ldots ,n\right)}\cup {\left\{{\tilde{H}}_{i}^{t}\right\}}_{\left(i=n+1,\ldots ,n_{t}\right)}$ and $n^{t}$ represent the total number of $D_{s}$ and $D_{t}$ data, respectively, where ${\tilde{H}}_{i}^{t}$ is the *i*-th collected CSI matrix ${\tilde{H}}_{i}^{t}$ from the target environment. It is observed that, CSI matrix ${\tilde{H}}_{i}^{t}$ of the target environment has target label $y_{i}^{t}$, where 1≤i≤n. However, there are no target labels for all ${\tilde{H}}_{i}^{t}$, where $n+1\leq i\leq n_{t}$. That is, the target labels ${\left\{y_{i}^{t}\right\}}_{\left(i=1,\ldots ,n\right)}$ are not available for training.

**S1.** 
The source domain and target domain are mapped into the same feature space. Let $S_{i}=\phi {\left\{{\tilde{H}}_{i}^{s}\right\}}_{i=1,\ldots ,n_{s}}$, $U_{j}=\phi {\left\{{\tilde{H}}_{i}^{t}\right\}}_{i=1,\ldots ,n}$, $L_{k}=\phi {\left\{{\tilde{H}}_{i}^{t}\right\}}_{i=n+1,\ldots ,n_{t}}$, then dot product is used to calculate the similarity of the source domain and the target domain. The similarity of the domain features is calculated by a similarity matrix between the source domain and the target domain, $F_{ij}=S_{i}\times U_{j}$ and $G_{ik}=S_{i}\times L_{k}$, where $F_{ij}$ is the similarity matrix of the unlabeled data of the source domain and the target domain, and $G_{ik}$ is the similarity matrix of the labeled data of the source domain and the target domain.**S2.** 
After obtaining the similarity matrix, $F_{ij}$, a conversion probability, $P_{ij}^{SU}$, of the source domain followed by [20] is
(23)$$\begin{array}{c} P_{ij}^{SU}=P\left(U_{j}|S_{i}\right)=SM_{columns}{\left(F\right)}_{ij}=\frac{\left(exp\left(F_{ij}\right)\right)}{\sum_{j^{\prime }}exp(F_{ij^{\prime }})} \end{array}$$
where $P_{ij}^{SU}$ is the conversion probability of the similarity matrix $F_{ij}$ by applying the softmax function for the column of $F_{ij}$, and the first row of $P_{ij}^{SU}$ is expressed as the probability of similarity of data ${\tilde{H}}_{i}^{s}$ to all unlabeled data in the target domain. For more consideration of DADA, we further calculate $P_{ik}^{SL}$ as follows. The *i*-th row of $P_{ik}^{SL}$ is expressed as the similarity probability of the source domain data ${\tilde{H}}_{i}^{s}$ to that of all labeled data of the target domain, where $1\leq i\leq n_{s}$. Similarly, the conversion probability from [20] of the target domain of $F_{ij}$ is,
(24)$$\begin{array}{c} P_{ij}^{US}=P\left(S_{i}|U_{j}\right)=SM_{rows}{\left(F\right)}_{ij}=\frac{\left(exp\left(F_{ij}\right)\right)}{\sum_{i^{\prime }}exp(F_{i^{\prime }j})} \end{array}$$
where $P_{ji}^{US}$ is the conversion probability of the similarity matrix $F_{ij}$ by applied the softmax function to the row of $F_{ij}$, the first row of $P_{ji}^{US}$ is expressed as the similarity probability of the data ${\tilde{H}}_{n+1}^{t}$ to all data of the source domain. For more consideration of DADA, we also further calculate $P_{ik}^{LS}$ as follows. The *i*-th row of $P_{ik}^{LS}$ is expressed as the similarity probability of the labeled data of the target domain ${\tilde{H}}_{i}^{t}$ to that of the source domain, where $n+1\leq i\leq n_{t}.$ Following [20], the subsequent calculation of the associated similarity for unlabeled data in the target domain can be expressed as:
(25)$$\begin{array}{c} P^{SUS}={\left(P^{SU}P^{US}\right)}_{ij}=\sum_{n}P_{in}^{SU}P_{ni}^{US} \end{array}$$
where $P_{ij}^{SUS}$ [20] is the round-trip probability of similarity matrix $F_{ij}$, starting from $S_{i}$ and ending at $S_{j}$. Assuming that the label mapped back to $S_{j}$ is unchanged relative to $S_{i}$, the label distribution of $S_{i}$ [20] is expressed as:
(26)$$\begin{array}{c} Y_{ij}{=\{}_{\;0\;\;\;\;\;else}^{1/S_{i}\;\;class\left(S_{i}\right)=class\left(S_{j}\right)} \end{array}$$The *i*-th column of $Y_{ij}$ [20] can be expressed as the probability of similarity between ${\tilde{H}}_{i}^{s}$ and other source domain data, and the cross-entropy with the round-trip probability can be expressed as:
(27)$$\begin{array}{c} L_{SUS}=H\left(Y_{ij},P^{SUS}\right) \end{array}$$
where $L_{SUS}$ is the difference degree function [20] quantified by cross-entropy, which is mainly mapped to the unlabeled data of the target domain through the label data of the source domain, and then is mapped back to the source domain and distributed with the label data of the source domain, to compare the degree of difference to quantify the distance between the two domains. However, this round-trip mapping cannot directly reflect the difference degree, so we further modify the ADA by dynamically utilizing a different ratio of labeled data of the target domain to map back to the source domain to obtain the difference degree. Since both parties have labels, the new defined cross-entropy calculation, $L_{LS}$, is performed through the conversion probability of $P_{ik}^{LS}$ and the distribution probability $J_{ij}$ of the label data of the target domain mapped to the source domain,
(28)$$\begin{array}{c} L_{LS}=H\left(J_{ij},P^{LS}\right) \end{array}$$The *i*-th row of $J_{ik}$ is the probability of similarity between ${\tilde{H}}_{i}^{t}$ and all labeled data of the source domain, where 1≤i≤n. Assuming that the label mapped to $S_{i}$ relative to $L_{k}$ is unchanged, the label distribution of $Y_{ij}$ can be expressed as,
(29)$$\begin{array}{c} J_{ik}=\left\{{}_{\;0\;\;\;\;\;else}^{\frac{1}{L_{k}}\;\;class\left(L_{k}\right)=class\left(S_{i}\right)}\right. \end{array}$$The divergence between the two domains is,
(30)$$\begin{array}{c} L_{div}=max\left[H\left(Y_{ij},P^{SUS}\right),H\left(J_{ij},P^{LS}\right)\right]=max\left[Q\left(Y_{ij},P^{SUS}\right),Q\left(J_{ij},P^{LS}\right)\right] \end{array}$$
where $L_{div}$ is the loss of the divergence of the two domains. Two different mapping functions are referenced to illustrate the distance degrees.**S3.** 
The difference loss function of two domains is only to correlate the simple and easily correlated data in the unlabeled target domain, a visit loss, followed by [20], is needed and given.
(31)$$\begin{array}{c} L_{vis}=Q\left(T,P^{vis}\right)\\ subject\;to\left\{\begin{array}{c} P_{j}^{vis}=\sum_{j^{\prime }}P_{ij^{\prime }}^{SU}\\ T_{j}=\frac{1}{|U_{j}|} \end{array}\right. \end{array}$$
where $P_{ij}^{SU}$ is calculated by adding up the columns in line units, and to calculate the cross-entropy with $T_{j}$. It is unreasonable that the calculation of $L_{vis}$ under the data distribution must be balanced [20]. This is because the number of unlabeled data is unknown before training. To provide the data imbalance capability and release the limitation of $L_{vis}$, our DADA scheme replaced traditional $L_{vis}$ [20] with a new loss function of synchronization, denoted $L_{syn}$, as follows:
(32)$$\begin{array}{c} L_{syn}=Q\left(P_{j}^{US},P_{j}^{SU}\right)\\ subject\;to\left\{\begin{array}{c} P_{j}^{SU}=\sum_{j^{\prime }}P_{ij^{\prime }}^{SU}\\ P_{j}^{US}=\sum_{j^{\prime }}SM_{columns}\left({P_{ij^{\prime }}^{SU}}^{T}\right) \end{array}\right. \end{array}$$
where $P_{j}^{SU}$ adds up the columns in line units and ${P_{ij}^{US}}^{T}$ adds up the columns in line units to make sure that both $P_{j}^{SU}$ and ${P_{ij}^{US}}^{T}$ are still kept in the same distribution. This work measured the correlation which can only avoid correlating the simple and easily correlated data in the unlabeled target domain, under the data distribution of $U_{j}$, which is imbalance. Finally, $L_{sim}\left(D_{s},D_{t}\right)$ is obtained by
(33)$$\begin{array}{c} L_{sim}\left(D_{s},D_{u},D_{l}\right)=\beta L_{dis}+\left(\beta -1\right)L_{syn}. \end{array}$$
where β is the hyper-parameter of the combined targets. $L_{sim}$ is represented as the combined loss, $L_{div}$ and new constructed $L_{syn}$.

As shown in Figure 11, the source domain $D_{s}={\left\{{\tilde{H}}_{i}^{s},y_{i}^{s}\right\}}_{\left(i=1,\ldots ,n_{s}\right)}$ and the target domain, $D_{t}={\left\{{\tilde{H}}_{i}^{t},y_{i}^{t}\right\}}_{\left(i=1,\ldots ,n\right)}\cup {\left\{{\tilde{H}}_{i}^{t}\right\}}_{\left(i=n+1,\ldots ,n_{t}\right)}$, are mapped into the same space using a DNN embedder, while the similarity matrix is calculated to have the association probability matrix process as shown in Figure 12, by obtaining $P^{SU}$,$P^{US}$, $P^{SL}$ and $P^{LS}$. The first row of $P^{SU}$ is the possibility of $H_{1}^{s}$ to all unlabeled data of the target domain. The first row of $P^{US}$ is the possibility of $H_{n+1}^{t}$ to all data of the source domain. The first row of $P^{SL}$ is the possibility of $H_{1}^{s}$ to all labeled data of the target domain. The first row of $P^{LS}$ is the possibility of $H_{1}^{t}$ to all data of the source domain. Figure 13 provides the calculating process of the divergence loss by calculating the cross-entropy between the possibility matrix and the true value matrix. The domain difference is calculated through the unlabeled data of the target domain and the data of the source domain. The key improvement of our DADA is that we additionally utilize the labeled data of the target domain for the association calculation. The first row of matrix *Y* is the truth value of $H_{1}^{s}$. By calculating the cross-entropy with the estimated value of $P^{SUS}$, the similarity score is obtained between data of the source domain and unlabeled data of the target domain. The last row of matrix *J* is the truth value of $H_{n_{t}}^{t}$. By calculating the cross-entropy with the estimated value of $P^{LS}$, the similarity score is also obtained. Figure 14 illustrates the calculating process of the synchronize loss, and synchronize loss is improved by the assistance of the visit loss proposed by ADA. The synchronize loss is calculated by the cross-entropy between $P_{j}^{US}$ and $P_{j}^{SU}$. It is noted that our proposed DADA can overcome the data imbalance issue of the target domain, while ADA is assumed that the data balance issue is required.

The time complexity of step 1 is $n_{s}\;\times \;n_{t}$, and the time complexity of step 2 is $n_{s}+n_{t}$, the time complexity of step 3 is $n_{s}$, where $n_{s}$ is the number of source data, and $n_{t}$ is the number of target data. Since the number of categories is a fixed value, the overall time complexity of Algorithm 2 is $O(n_{s}\;\times \;n_{t})$.
**Algorithm 2:** The dynamic associate domain adaptation phase.**Input**:The source domain label data $D_{s}={\left\{{\tilde{H}}_{i}^{s},y_{i}^{s}\right\}}_{i=1\ldots n_{s}}$, the target domain label data $D_{t}={\left\{{\tilde{H}}_{i}^{t},y_{i}^{t}\right\}}_{i=1\ldots n}$ and the target domain unlabelled data $D_{t}={\left\{{\tilde{H}}_{i}^{t}\right\}}_{i=n+1\ldots n_{t}}$.
**Output**:Similarity loss
$L_{sim}$.- - - - - - - - - - - - - - - - - - - - - - - Step 1 - - - - - - - - - - - - - - - - - - - - - -Mapped the data to same space by ϕ.$S_{i}=\phi \left({\left\{{\tilde{H}}_{i}^{s}\right\}}_{i=1\ldots n_{s}}\right)$, $U_{j}=\phi \left({\left\{{\tilde{H}}_{i}^{t}\right\}}_{i=1\ldots n}\right)$, $L_{k}=\phi \left({\left\{{\tilde{H}}_{i}^{t}\right\}}_{i=n+1\ldots n_{t}}\right)$Calculate similarity matrix $F_{ij}$ and $G_{ik}$ by dot product.$F_{ij}=S_{i}\times U_{j}$, $G_{ik}=S_{i}\times L_{k}$Calculate conversion probability matrix of the similarity matrix $F_{ij}$ and $G_{ik}$ by softmax function SM.$P_{ij}^{SU}=P\left(U_{j}|S_{i}\right)=SM_{columns}{\left(F\right)}_{ij}=\frac{exp(F_{ij})}{\Sigma_{j^{\prime }}exp\left(F_{ij^{\prime }}\right)}$$P_{ji}^{US}=P\left(S_{i}|U_{j}\right)=SM_{rows}{\left(F\right)}_{ij}=\frac{exp(F_{ij})}{\Sigma_{i^{\prime }}exp\left(F_{i^{\prime }j}\right)}$$P_{ik}^{SL}=P\left(L_{k}|S_{i}\right)=SM_{columns}{\left(G\right)}_{ik}=\frac{exp(F_{ik})}{\Sigma_{k^{\prime }}exp\left(F_{ik^{\prime }}\right)}$$P_{ki}^{LS}=P\left(L_{k}|U_{j}\right)=SM_{rows}{\left(G\right)}_{ik}=\frac{exp(F_{ik})}{\Sigma_{i^{\prime }}exp\left(F_{i^{\prime }k}\right)}$- - - - - - - - - - - - - - - - - - - - - - - Step 2 - - - - - - - - - - - - - - - - - - - - - -Calculate $L_{SUS}$ by conversion probability,$P^{SUS}$ is the round-trip probability and the label distribution $Y_{ij}$.$P^{SUS}={\left(P^{SU}P^{US}\right)}_{ij}=\Sigma_{n}P_{in}^{SU}P_{ni}^{US}$$Y_{ij}{=\{}_{\;0\;\;\;\;\;else}^{\frac{1}{S_{i}}\;\;class\left(S_{i}\right)=class\left(S_{j}\right)}$$L_{SUS}=Q\left(Y_{ij},P^{SUS}\right)$ Calculate $L_{LS}$ by conversion probability, $P^{LS}$ is the conversion probability and the label distribution $J_{iK}$.$J_{ik}{=\{}_{\;0\;\;\;\;\;else}^{\frac{1}{L_{k}}\;\;class\left(L_{k}\right)=class\left(S_{i}\right)}$$L_{LS}=H\left(J_{ij},P^{LS}\right)$- - - - - - - - - - - - - - - - - - - - - - - Step 3 - - - - - - - - - - - - - - - - - - - - - -Combine $L_{SUS}$ and $L_{LS}$ as divergence loss $L_{div}$.$L_{div}=max\left[L_{SUS},L_{LS}\right]$Calculate synchronize loss $L_{syn}$.$L_{syn}=H\left(P_{j}^{US},P_{j}^{SU}\right)$$P_{j}^{SU}=\Sigma_{j^{\prime }}P_{ij^{\prime }}^{SU}$$P_{j}^{US}=\Sigma_{j^{\prime }}{P_{ij^{\prime }}^{US}}^{T}$$L_{syn}=Q\left(P_{j}^{US},P_{j}^{SU}\right)$Combine $L_{div}$ and $L_{syn}$ as $L_{sim}$.$L_{sim}=\beta L_{div}+\left(\beta -1\right)L_{syn}$

### 4.4. Associate Knowledge Fine-Tuning Phase

In the last phase, the learned features are transferred through the HAR of the image with domain-invariant characteristics, and the shallow weights of the source domain learned through the pre-training phase are frozen as common features, and knowledge transfer is performed on the deep layer of the model. The combined loss $L_{sim}$ of the dynamic associate domain adaptation phase is used to fit the feature distributions of the two domains as follows.

**S1.** 
The labeled data of the source domain and both of the labeled and unlabeled data of the target domain are trained simultaneously. To preserve the features learned in the pre-training phase, the stable layers are frozen, and the output, before the flattening layer, is expressed as $z_{o}^{P}$. The maximum pooling operation MP is applied to the feature to extract the maximum near-row flattening. This operation is expressed as:
(34)$$\begin{array}{c} m_{o}=MP\left(z_{o}^{p}\right) \end{array}$$
where $m_{o}$ is the set of data features from the source domain $D_{s}$ and domain domain $D_{t}$, and is used as a flattened feature, and $m_{o}$ passes through the *k*-layer fully connected layer to calculate the similarity of the *k*-layer feature $L_{sim}$, given as:
(35)$$\begin{array}{c} \sum_{f_{k}}^{l=f_{1}}L_{sim}\left(D_{s}^{l},D_{t}^{l}\right) \end{array}$$
where *l* is the current number of layers. The similarity values of the *k*-layer features are accumulated as part of the loss.**S2.** 
The final feature is obtained by $d=d^{s}\cup d^{t}$, which is used for the final activity prediction through the fully connected layer with the activation function of softmax, where $p_{i}^{s}=W_{o}\times d^{s}+b_{o}$, $p_{i}^{t}=W_{o}\times d^{t}+b_{o}$, where $W_{o}$ and $b_{o}$ are trainable parameters, and $p_{i}^{s}$ is the predicted value of ${\tilde{H}}_{i}^{s}$, and $p_{i}^{t}$ is the predicted value of ${\tilde{H}}_{i}^{t}$. Finally, the total loss can be represented as:
(36)$$\begin{array}{c} arg\;\times \;min\;L_{f}\\ L_{f}=\lambda L_{c}+\left(1-\lambda \right)\sum_{l=f_{1}}^{f_{k}}L_{sim}\left(D_{s}^{l},D_{t}^{l}\right)\\ =\lambda \times max\left[\frac{1}{n_{s}}\sum_{i=1}^{n_{s}}Q\left(y_{i}^{s},p_{i}^{s}\right),\frac{1}{n}\sum_{i=1}^{n}Q\left(y_{i}^{t},p_{i}^{t}\right)\right]+\left(1-\lambda \right)\sum_{l=f_{1}}^{f_{k}}L_{sim}\left(D_{s}^{l},D_{t}^{l}\right) \end{array}$$
where $Q\left(y_{i}^{s},p_{i}^{s}\right)$ and $Q\left(y_{i}^{t},p_{i}^{t}\right)$ are used to solve the classification of the source domain and the target domain, respectively. Note that $n_{s}$ and *n* are the numbers of labeled data of the source domain and the target domain, respectively, *m* is the number of classification categories, $y_{i,j}^{s}$ and $y_{i,j}^{t}$ are denoted as the *i*-th labeled data in $D_{s}$ and $D_{t}$, respectively, and the data should belong to *j*-th category, and $p_{i,j}^{s}$ and $p_{i,j}^{t}$ are the predicted probabilities of $D_{s}$ and $D_{t}$, respectively. Let $\sum_{l=f_{1}}^{f_{k}}L_{sim}\left(D_{s}^{l},D_{t}^{l}\right)$ be the similarity between two domains of the fully connected layer. The final goal is to minimize $L_{f}$, which $L_{f}$ is the combined function of $L_{sim}$ and $L_{c}$, where λ is the hyper-parameter of the hybrid objective function, which 0≤λ≤1.

As shown in Figure 15, the weights before the flatten layer are frozen and the similarity loss with all the data of both domains are calculated. The backpropagation operation is done to update the weights of the fully connected layer. For instance, as shown in Figure 15, $\sum_{l=f_{1}}^{f_{3}}L_{sim}\left(D_{s}^{l},D_{t}^{l}\right)$ has similarity loss under $max\left[\frac{1}{n_{s}}\sum_{i=1}^{n_{s}}Q\left(y_{i}^{s},p_{i}^{s}\right),\frac{1}{n}\sum_{i=1}^{n}Q\left(y_{i}^{t},p_{i}^{t}\right)\right]$, where $f_{3}$ denotes an example of three fully connected layers in our work.

The time complexity of step 1 is $n_{s}\;\times \;n_{t}$, and the time complexity of step 2 is $n_{s}+n_{t}$, where $n_{s}$ and $n_{t}$ are the numbers of data of the source and target domains. The the number of categories is a fixed value, and the overall time complexity in Algorithm 3 is $O(n_{s}\;\times \;n_{t})$.
**Algorithm 3:** The associate knowledge fine-tuning phase.
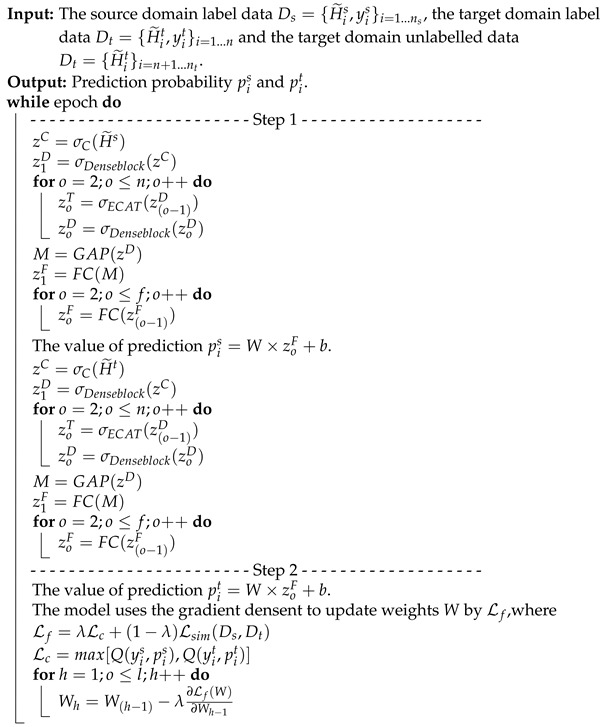


## 5. Experimental Result

The experimental setup is described in Section 5.1 and the performance evaluation is then discussed in Section 5.2.

### 5.1. Experimental Setup

The environment setup is described, mainly including the model parameter settings for our experimental results. In our experimental, the AI GPU utilizes NVIDIA RTX 3080, and the AI framework adopts pytorch version 1.4.0. The programming environment is the Python 3.8 version under Windows 10. The WiFi CSI data acquisition framework uses the Intel IWL 5300 NIC tool [27], two computers installed with Ubuntu 14.04.4 equipped with the Intel 5300 NIC are used as the interface programming environment. The recognition activity patterns in our experimental are: jumping, squatting, sitting, standing, and falling. The experimental parameters are given in Table 1, and the data size of each data is 90×3000 pixel. In the source domain, as shown in Figure 16a where the size of the source environment is 5 m × 4 m, we have collected 5000 data, and is divided into a 4000 training set and a 1000 testing set. Two different sizes of target domains are shown, denoted as target domain A with sizes of 10 m × 4 m, as shown in Figure 16b and target domain B with sizes of 3 m × 4 m, as illustrated in Figure 16c; each one had collected 2000 data, and is divided into a 1500 training set and a 500 testing set. We collected 5000 data from the source domain and 4000 data from two target domains, which are obtained by 11 participants in the laboratory, with a height of about 160 to 180 cm and a weight of about 65 kg to 100 kg.

In the experiment, we use source, target (A), and target (B) to represent the source domain, target domain A, and target domain B. The data augmented dataset is 1.5 times than that of the original training set by adopting the rotate, map, and mask techniques. To illustrate the effect of environment-independent human activity recognition, the source domain is used to train the pre-trained model, and the pre-trained model is used to transfer trained knowledge to target (A) and target (B). The experimental environment with three pairs of WiFi transmitter and receivers is illustrated in Figure 17.

There are many recent works of environment-dependent HAR. In general, the distance between the transmitter and receivers as it increases, the lower human activity recognition accuracy will be. In our work, we mainly investigate the problem of environment-independent HAR. To more easily understand the effects and our contribution of environment-independent HAR, we fixed some parameters; the number of WiFi transmitters and receivers and distance between the transmitter and receivers when moving to the new target domains, and only focus on the parameters under consideration of the different ratios of labeled/unlabled datasets and under the consideration of the data imbalance problem are as follows.

To investigate the effect of the data imbalance problem, we use various data number ratios of the target domain, denoted as target data ratio and represented as number of training data of squat pattern: number of training data of sitting pattern: number of training data of stand pattern: number of training data of jump pattern: number of training data of fall pattern. In our experiment, three kinds of target data ratio; (1:1:1:1:1), (1:1:0.5:1:0.5), and (1:1:1:1:0.1), are considered. Specially, there is no data imbalance condition if target data ratio (1:1:1:1:1) is assumed. The data imbalance issue is investigated if target data ratios (1:1:0.5:1:0.5) and (1:1:1:1:0.1) are assumed.

To investigate the effect of the dynamic ADA issue, the ratios of labeled data to the unlabeled data of the target domains, target (A), and target (B) are 0%, 25%, 50%, 70%, and 100%, where the ratio value = 0% represented all of the data of unlabeled target domains, and the ratio value = 100% represented all of the data of labeled target domains. Therefore, we will have 15 combination cases under various target data ratios and various ratios of labeled data.

### 5.2. Performance Analysis

This subsection discussed the performance analysis, and the performance metrics of our experiment to be observed are:(1)**Pre-training accuracy (PTA)** is the recognition accuracy to predict the correct HAR pattern from five kinds of HAR patterns in the same source environment, while the pre-trained model is trained from a source domain.(2)**Recognition accuracy (RA)** is the recognition accuracy to predict the correct HAR pattern from five kinds of HAR patterns in the new target environment, under a given pre-trained model that is trained from a source domain, which is quite different from the source environment.(3)**Time cost (TC)** is the total time cost, which is the sum of the processing time, pre-training time, and fine-tuning time.

An efficient HAR scheme using WiFi CSI signals is achieved with a high PTA, high RA, and low TC. Efforts will be made in this paper to improve the PTA, RA, and reduce the TC.

#### 5.2.1. Pre-Training Accuracy (PTA)

The experimental results of the pre-training accuracy (PTA) vs. epochs are shown in Figure 18. The PTA is the ratio of the number of correct classification prediction to the total number of predictions. In Figure 18a, the solid line represents the prediction with the data augmentation, and the dashed line represents the prediction without data augmentation. The data augmentation expanded the training data 1.5 times. In the pre-training result, the red line represents as our attention-based DenseNet (AD) scheme, the blue line represents the HAR-MN-EF scheme [19], the green line represents the DANDR scheme [25], and the orange line represents the WiLlSensing scheme [24]. Figure 18a illustrates that the PTA of AD is better than that of other schemes due to the advantage of the feature attention and reuse strategy. The DANGR scheme does not use the feature enhancement, which leads to poor learning efficiency. All of the other schemes adopt the feature enhancement, but lead to the huge differences in PTA. In general, feature enhancement is helpful for pre-training. The experimental result also shows that the PTA of pre-training schemes with data augmentation and attention mechanisms is higher than that of pre-training schemes without data augmentation and an attention mechanism.

The PTA of pre-training with the different number of antennas vs. epoch is illustrated in Figure 18b. The number of antennas affects PTA. The greater the number of antennas, the higher PTA will be. With the same number of antennas, the PTA of our AD scheme is better than that of other schemes. It supports that the feature extraction of our proposed AD scheme is also better than that of other schemes. Table 2 and Table 3 show the confusion matrices of our proposed AD scheme and HAR-MN-EF scheme [19]. The result shows that our proposed AD scheme actually improves the accuracy.

#### 5.2.2. Recognition Accuracy (RA)

Given the pre-training from the source domain, the performance results of RA for target (A) and target (B) under the various ratios of labeled data = 0%, 25%, and 100% vs. epoch are given in Figure 19. Initially, if the target data ratio is (1:1:1:1:1), the proportion of each activity in the training dataset is equal. When the pre-training knowledge is transferred to target (A) or target (B), the weight before the flattening layer is frozen as a common feature. Therefore, RA of the 0th epoch is not 0. The accuracy of the pre-training knowledge affects RA results, our pre-training knowledge adopting the AD model is higher than other models. Figure 19a,b show the performance of RA vs. epoch under ratios of labeled data = 0% and target data ratio = (1:1:1:1:1) for moving from the source domain to target (A) and target (B). To make a comparison, when we apply the AD scheme for some models as the same pre-training model, we observed that the average RA of our AD-DADA scheme > that of AD-ADA [20] scheme > that of AD-MK-MMD [25] scheme > that of AD-MMD [32] scheme. In addition, the average RA of our AD-DADA scheme > that of DANDR [25] scheme > that of HAR-MN-EF [19] scheme from the perspective of epoch. Figure 19c,d show the performance of RA vs. epoch under ratios of labeled data = 25% and target data ratio = (1:1:1:1:1) for moving from the source domain to target (A) and target (B). We observed that the average RA of our AD-DADA scheme > that of AD-ADA scheme > that of AD-MK-MMD scheme > that of AD-MMD scheme > that of DANDR scheme > that of HAR-MN-EF scheme from the perspective of epoch. Figure 19e,f show the performance of RA vs. epoch under ratios of labeled data = 100% and target data ratio = (1:1:1:1:1) for moving from source domain to target (A) and target (B). We observed that the average RA of our AD-DADA scheme > that of AD-ADA scheme > that of AD-MK-MMD scheme > that of AD-MMD scheme > that of DANDR scheme > that of HAR-MN-EF scheme from the perspective of epoch. In addition, we also investigate RA under various ratios of labeled data = 50% and 70% in Table 4. Figure 20 shows RA vs. various ratios of labeled data for (a) target (A) and (b) target (B) under target data ratio = (1:1:1:1:1). In general, the higher the ratio of labeled data is, the higher RA is. We also observed that the average RA of our AD-DADA scheme > that of AD-ADA scheme > that of AD-MK-MMD scheme > that of AD-MMD scheme > that of DANDR scheme > that of HAR-MN-EF scheme from the perspective of ratios of labeled data. The improvement of RA for target (B) is better than that of target (A). This is because target (B) is a small area.

The performance of RA vs. epoch is given under ratios of labeled data = 0% (Figure 21a,b), ratios of labeled data = 25% (Figure 21c,d), and 100% (Figure 21e,f), under the target data ratio sets to (1:1:0.5:1:0.5) for target (A) and target (B). For the data imbalance of the target domain, we have the general results that the RA of our AD-DADA scheme > that of AD-ADA scheme > that of AD-MK-MMD scheme > that of AD-MMD scheme > that of DANDR scheme > that of HAR-MN-EF scheme from the perspective of epoch. This is because the ADA scheme is limited by its balance distribution of the target domain, and DADA, MK-MMD, and MMD schemes are not subject to this limitation. Figure 22 provides the experimental result of RA vs. various ratios of labeled data for (a) target (A) and (b) target (B) under target data ratio = (1:1:0.5:1:0.5). Figure 23 offers the experimental result of RA vs. various ratios of labeled data for (a) target (A) and (b) target (B) under target data ratio = (1:1:1:1:0.1). Similarly, we have the general results that the RA of our AD-DADA scheme > that of AD-ADA scheme > that of AD-MK-MMD scheme > that of AD-MMD scheme > that of DANDR scheme > that of HAR-MN-EF scheme from the perspective of various ratios of labeled data. In addition, we also provide recall and precision, as shown in Figure 24, under various target data ratios for target (A) and target (B). We observed when the target data ratio is more imbalanced, the recall is higher than the precision, which means that a class with data imbalance will be more easily judged as other classes.

To illustrate the effect of data augmentation, Table 4 provides the RA result with data augmentation, and Table 5 offers the RA result without data augmentation when performing the associate knowledge fine-tuning phase. In general, the RA of the associate knowledge fine-tuning phase without data augmentation is higher than that of the associate knowledge fine-tuning phase with data augmentation. It is not useful to adopt the data augmentation technique for the target domain when executing the associate knowledge fine-tuning phase. Consequently, Table 6 summarizes all of the RA results of AD-DADA, AD-ADA, AD-MK-MMD, and AD-MMD schemes under the various target data ratio sets, and the various ratios of labeled data for target (A) and target (B). Under a fixed ratio of labeled data, the greater the data imbalance, the low the value of RA will be. Under a fixed target data ratio set, the greater the ratio of labeled data, the higher the value of RA will be. For instance, we observed that if the ratio of labeled data = 25%, the RA can be improved, especially if the data imbalance problem has occurred. In general, our proposed AD-DADA can provide a general adjustment scheme to dynamically increase the ratio of labeled data if a user is encountering a new poor target environment.

#### 5.2.3. Time Cost (TC)

The experimental results of the time cost (TC) of AD-DADA, AD-ADA, AD-MK-MMD, AD-MMD, DANGR, and WiLlSensing schemes are shown in Table 7. The total time cost is the sum of the processing time, pre-training time, and fine-tuning time, where the processing time is the time required in the data collection and processing phase. The pre-training time is the time required per epoch in the pre-training phase. The fine-tuning time is the time required per epoch in the fine-tuning phase. In general, the AD-based model utilizes the feature reuse strategy, so the time cost is higher than that of the DANGR and WiLlSensing schemes, as illustrated in Table 7. However, our AD-DADA scheme has the least time cost for all AD-based schemes, as shown in Table 7.

## 6. Conclusions

This paper addresses the problem of recognizing human activity independent of the environment, also known as domain adaptation. This work uses the channel state information (CSI) of WiFi signals. We have proposed semi-supervised transfer learning with dynamic adaptation of the associate domain in order to recognize human activity. To improve the recognition accuracy at the data pre-processing stage, missing packet filling, noise removal, background estimation, feature extraction, and feature enhancement are performed. The pre-trained model is trained from the source domain by collecting a complete set of labeled data for all of the human activity patterns of the CSI. The pre-trained model is then transferred to the target environment through the semi-supervised transfer learning stage. We proposed an algorithm for dynamically adapting the associated domain called DADA. The advantage of DADA is that it provides a dynamic strategy to remove different effects in different environments. An attention-based DenseNet (AD) model is developed as a training network, which is modified from the existing DenseNet by adding the attention feature. The proposed solution (DADA-AD) has been tested for five types of human activities (falling, standing, squatting, jumping, and sitting). The experimental results illustrate that the recognition accuracy of these activities in the test environment is 97.4%.

The automatic recognition of human activities is an important area for providing personalized care, mainly through human–-computer interaction analysis in medicine and sociology, to achieve a better cost performance, making it easy to implement in the actual architectural environment. The five types of human activities considered in this paper are a coarse-grained human activity model, not a fine-grained human activity model. The disadvantage is that the proposed WLAN sensing scheme is not suitable for recognizing fine-grained HAR. In addition, the setup locations of the transmitter and receiver of the WLAN-sensing solution are different, and the measured distance is limited. This limits the HAR applications and the implementation in a real building environment, under the requirement of a low cost–performance ratio.

To recognize the fine-grained human activity patterns, future work will extend these research experiences to an mmWave sensor network for automatic recognition of human activities with cost-effective and easy installation in a real building environment.

## Figures and Tables

**Figure 1 sensors-21-08475-f001:**
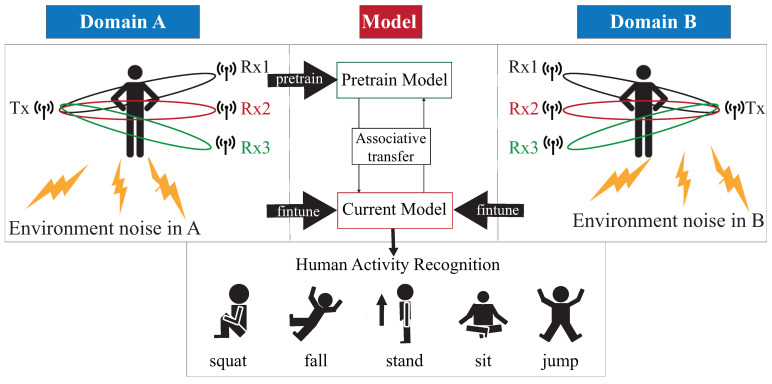
Our proposed DADA-AD scheme.

**Figure 2 sensors-21-08475-f002:**
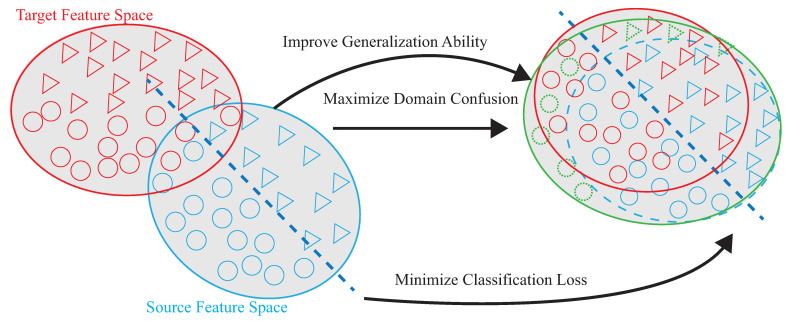
DADA-AD scheme improves generalization ability, maximize domain confusion, and minimize classification loss for source and target domains.

**Figure 3 sensors-21-08475-f003:**
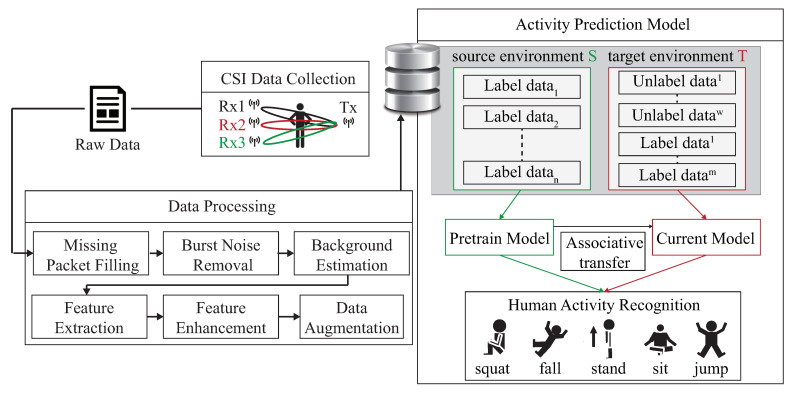
System structure of DA-DADAscheme.

**Figure 4 sensors-21-08475-f004:**
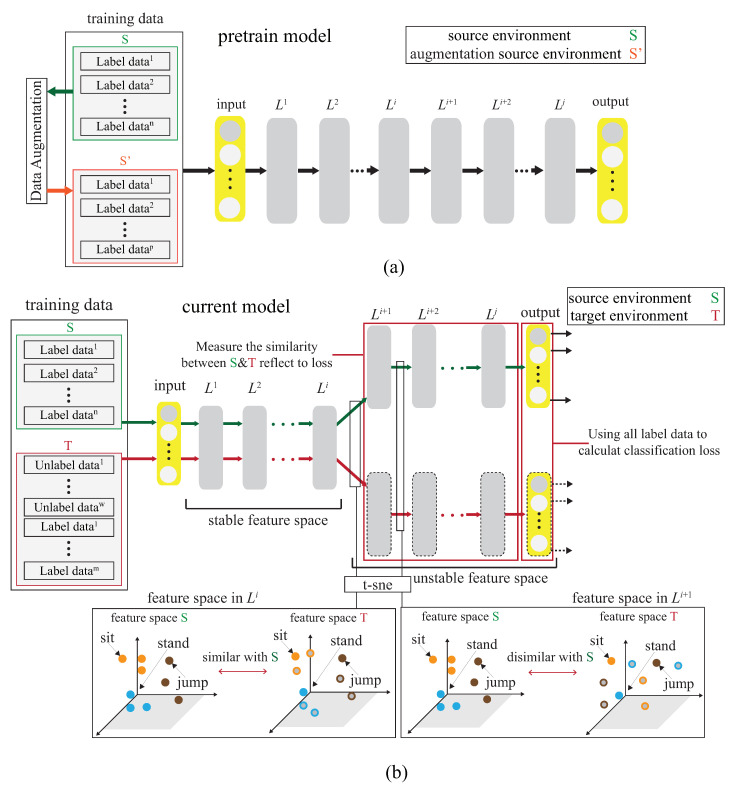
Basic idea of DA-DADA scheme. (**a**) The pre-trained process. (**b**) The fine-tuning process.

**Figure 5 sensors-21-08475-f005:**
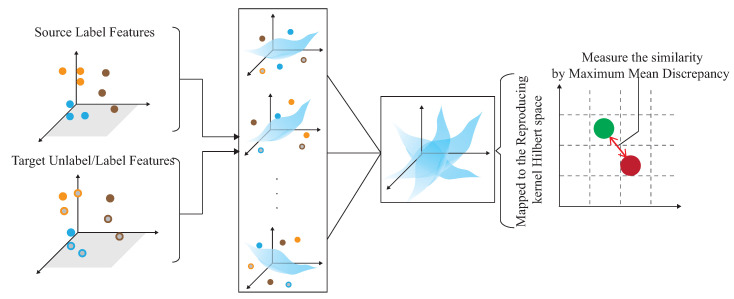
Transfer learning with minimized multiple kernels.

**Figure 6 sensors-21-08475-f006:**
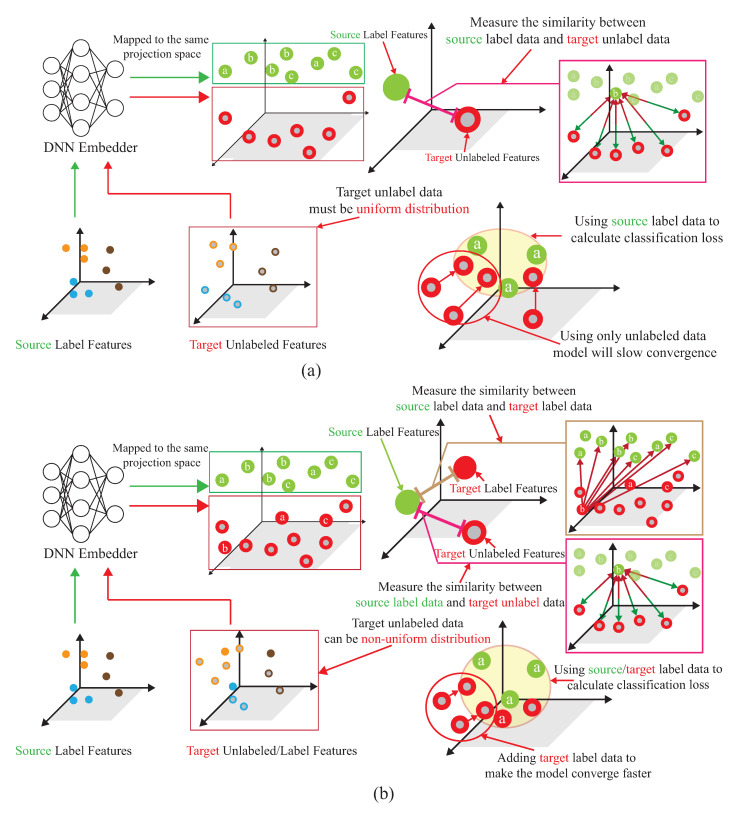
Comparison with (**a**) ADA scheme, and (**b**) our proposed AD-DADA scheme.

**Figure 7 sensors-21-08475-f007:**
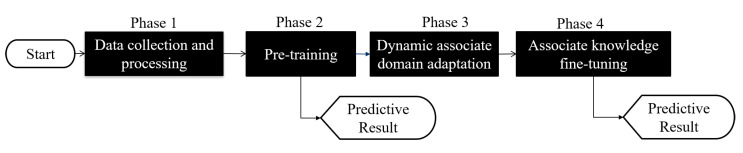
The flow chart of AD-DADA Scheme.

**Figure 8 sensors-21-08475-f008:**
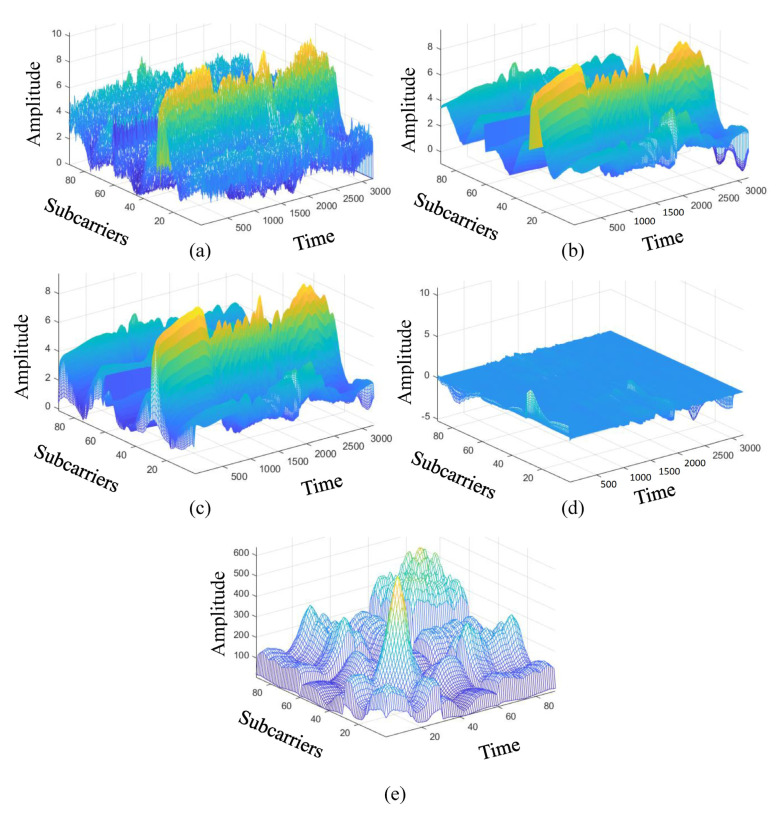
The 3-D diagrams of data pre-processing for the fall activity patterna, where (**a**) $H^{pf}$ matrix of missing packet filling operation, (**b**) denoising CSI matrix $H^{nr}$ of burst noise removal operation, (**c**) static CSI matrix $H^{be}$ of background estimation operation, (**d**) dynamic CSI matrix $H^{fe}$ of feature enhancement operation, and (**e**) the correlation matrix $\tilde{H}$ of feature enhancement operation.

**Figure 9 sensors-21-08475-f009:**
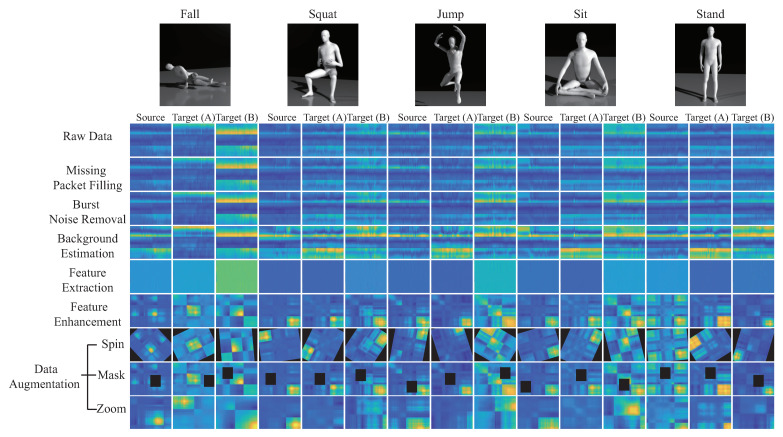
The 2D diagrams of CSI data pre-processing.

**Figure 10 sensors-21-08475-f010:**
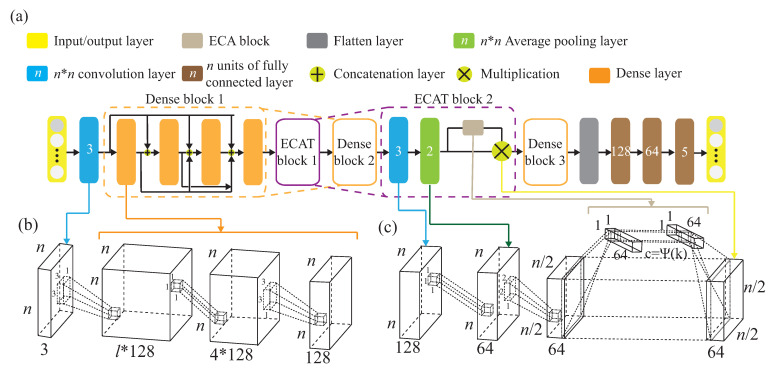
(**a**) The attention-based DenseNet (AD) model. (**b**) Illustration of denseblock with a 1×1 convolution layer and a 3×3 convolution layer. (**c**) Illustration of transition block with a 1×1 convolution layer, a 2×2 pooling layer, and an ECA structure.

**Figure 11 sensors-21-08475-f011:**
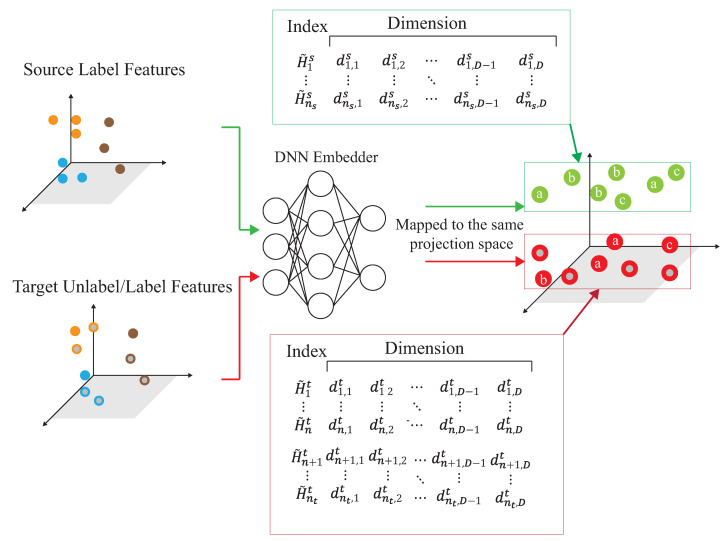
The mapping process to a projection space.

**Figure 12 sensors-21-08475-f012:**
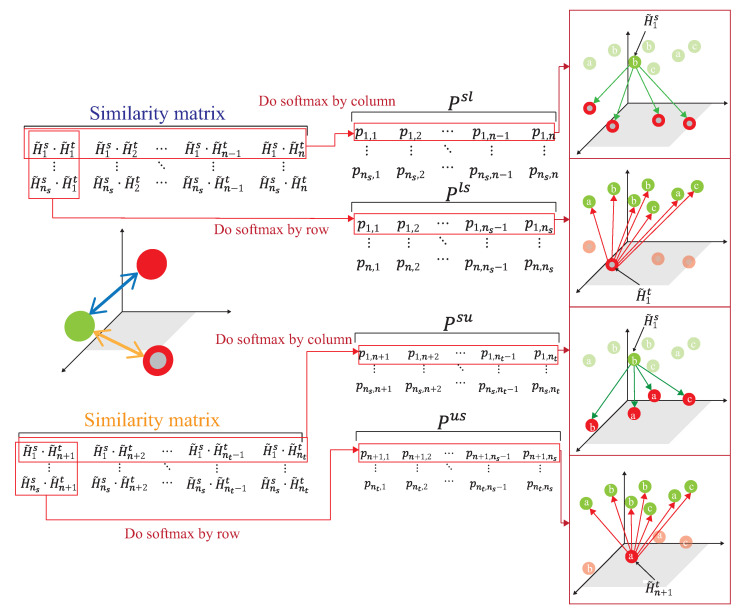
The association between source and target domains.

**Figure 13 sensors-21-08475-f013:**
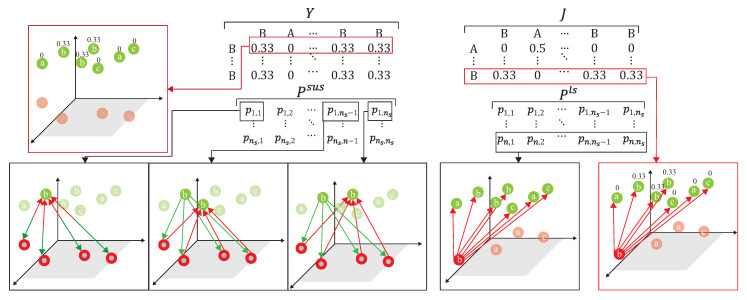
The process of divergence loss.

**Figure 14 sensors-21-08475-f014:**
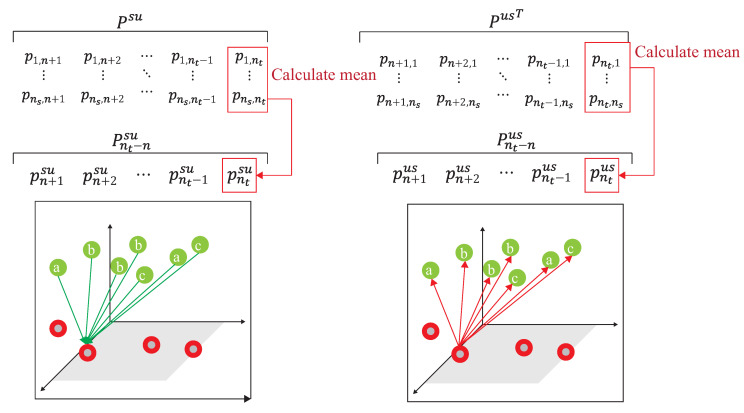
The process of synchronize loss.

**Figure 15 sensors-21-08475-f015:**
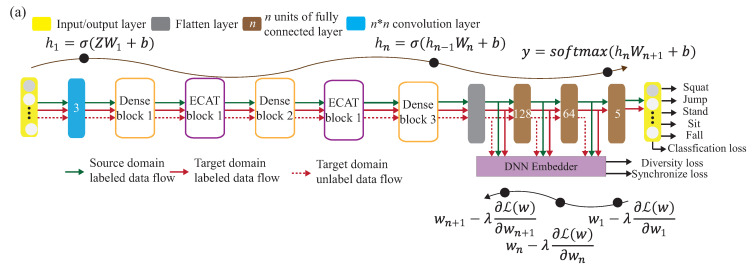
Example of the associate knowledge fine-tuning phase.

**Figure 16 sensors-21-08475-f016:**
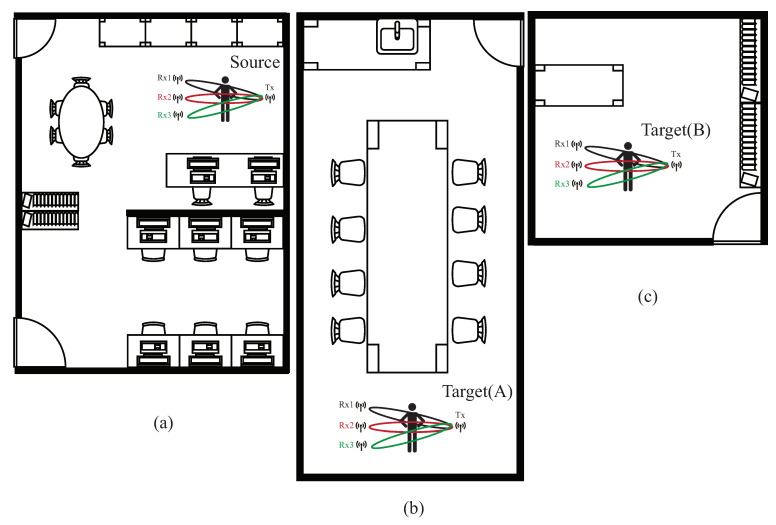
Layout of three experimental areas; (**a**) the source domain with the size of 5 m × 4 m, (**b**) the target domain A with big size of 10 m × 4 m, (**c**) the target domain B with the small size of 3 m × 4 m.

**Figure 17 sensors-21-08475-f017:**
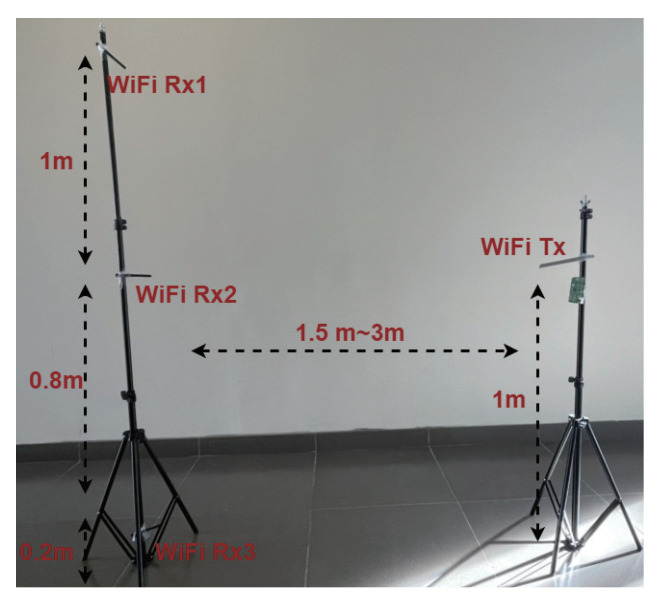
The experimental environment with three pairs of WiFi transmitter and receivers.

**Figure 18 sensors-21-08475-f018:**
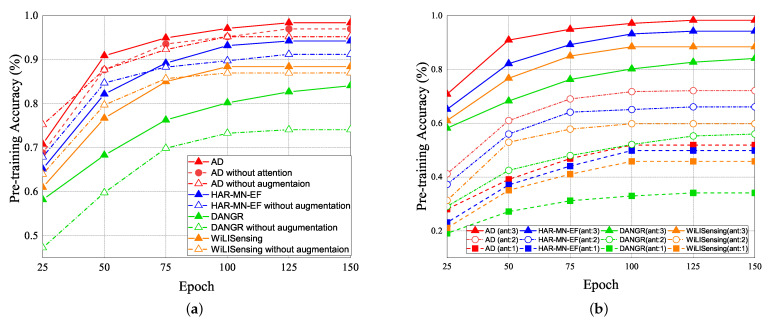
PTA vs. epoch for (**a**) data argumentation in different methods. (**b**) different number of antennas in different methods.

**Figure 19 sensors-21-08475-f019:**
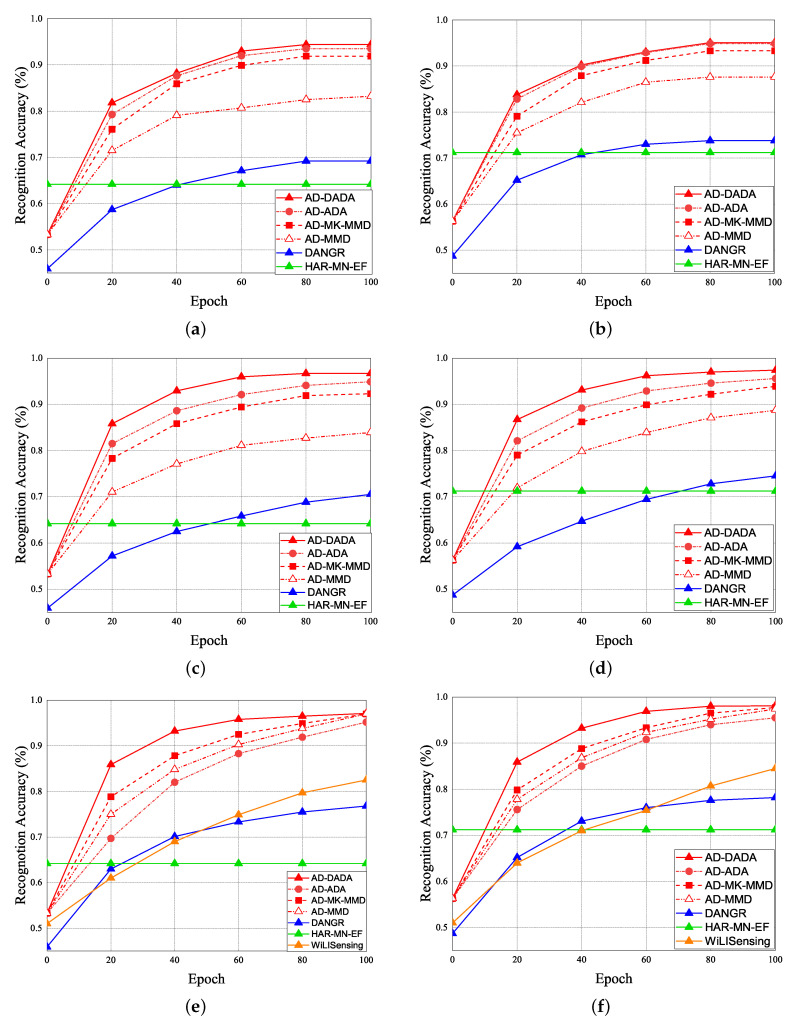
RA vs. epoch under ratios of labeled data = 0% and target data ratio = (1:1:1:1:1) for (**a**) target (A) and (**b**) target (B). RA vs. epoch under ratios of labeled data = 25% and target data ratio = (1:1:1:1:1) for (**c**) target (A) and (**d**) target (B). RA vs. epoch under ratios of labeled data = 100% and target data ratio = (1:1:1:1:1) for (**e**) target (A) and (**f**) target (B).

**Figure 20 sensors-21-08475-f020:**
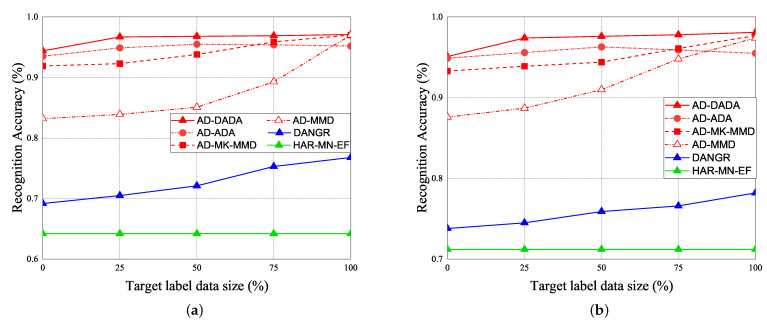
RA vs. various ratios of labeled data for (**a**) target (A) and (**b**) target (B) under target data ratio = (1:1:1:1:1).

**Figure 21 sensors-21-08475-f021:**
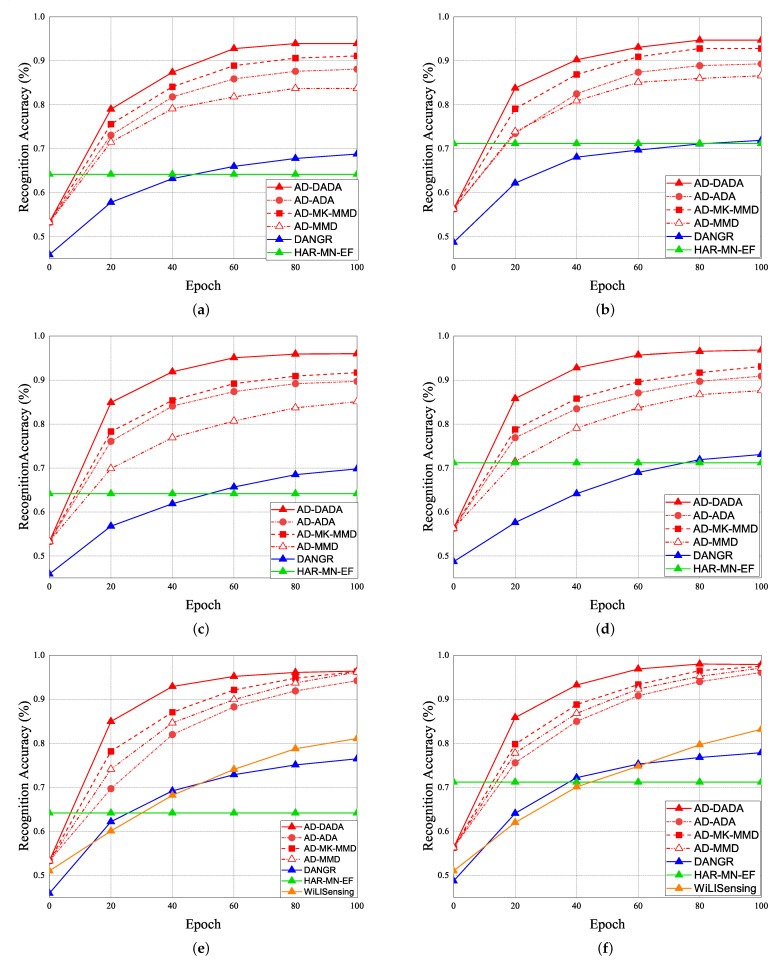
RA vs. epoch under ratios of labeled data = 0% and target data ratio = (1:1:0.5:1:0.5) for (**a**) target (A) and (**b**) target (B). RA vs. epoch under ratios of labeled data = 25% and target data ratio = (1:1:0.5:1:0.5) for (**c**) target (A) and (**d**) target (B). RA vs. epoch under ratios of labeled data = 100% and target data ratio = (1:1:0.5:1:0.5) for (**e**) target (A) and (**f**) target (B).

**Figure 22 sensors-21-08475-f022:**
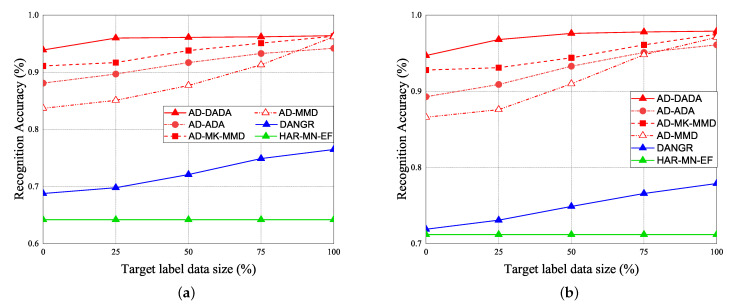
RA vs. various ratios of labeled data for (**a**) target (A) and (**b**) target (B) under target data ratio = (1:1:0.5:1:0.5).

**Figure 23 sensors-21-08475-f023:**
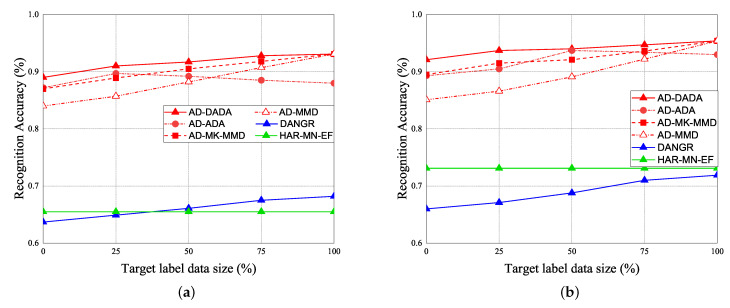
RA vs. various ratios of labeled data for (**a**) target (A) and (**b**) target (B) under target data ratio = (1:1:1:1:0.1).

**Figure 24 sensors-21-08475-f024:**
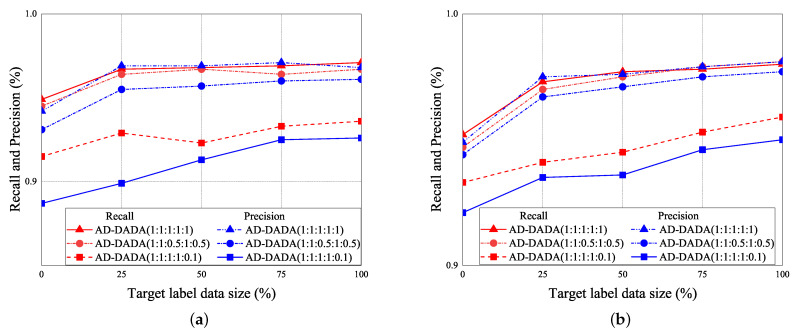
Recall and precision vs. various ratios of labeled data for (**a**) target (A) and (**b**) target (B) for AD-DADA scheme.

**Table 1 sensors-21-08475-t001:** Experiment parameter.

Environment	Source	Target (A)	Target (B)
Sampling frequency	1000 Hz		
Transmit antenna	1 antenna		
Receiving antenna	3 antenna		
Sampling time	3 s		
Subcarriers per link	30 subcarriers		
Source dataset	4000	1500	
Target dataset	1500	500	
Data expansion factor	1.5		
Weight adjustment λ	0.5		
Compression factor θ	0.5		
Weight adjustment β	0.5		
Learning rate	0.001		
Drop out	0.5		

**Table 2 sensors-21-08475-t002:** Confusion matrix of AD.

	Jump	Stand	Sit	Squat	Fall
**Jump**	1	0	0	0	0
**Stand**	0	0.97	0.01	0.02	0
**Sit**	0	0	0.99	0	0.01
**Squat**	0	0.01	0.02	0.965	0.005
**Fall**	0.005	0	0	0	0.995

**Table 3 sensors-21-08475-t003:** Confusion matrix of HAR-AF-DLN.

	Jump	Stand	Sit	Squat	Fall
**Jump**	0.995	0	0	0	0.005
**Stand**	0.005	0.92	0.04	0.03	0.005
**Sit**	0.005	0.01	0.98	0.005	0
**Squat**	0	0.085	0.02	0.885	0.01
**Fall**	0.005	0.005	0	0	0.99

**Table 4 sensors-21-08475-t004:** Recognition accuracy (RA) with data augmentation.

Model	Schemes	Target (A)		Target (B)	
		0%	25%	0%	25%
AD	DADA	88.7%	93.1%	89.9%	95.2%
	ADA	87.8%	90.4%	87.9%	91.9%
	MK-MMD	85.3%	88.4%	86.2%	89.9%
	MMD	76.9%	80.1%	80.8%	84.0%

**Table 5 sensors-21-08475-t005:** Recognition accuracy (RA) without data augmentation.

Model	Schemes	Target (A)		Target (B)	
		0%	25%	0%	25%
AD	DADA	94.4%	96.7%	95.1%	97.4%
	ADA	93.5%	94.9%	94.8%	95.6%
	MK-MMD	91.9%	92.3%	93.3%	93.9%
	MMD	83.2%	83.9%	87.6%	88.7%

**Table 6 sensors-21-08475-t006:** Recognition accuracy (RA) of all cases for encountering for target (A) and target (B).

Model	Target Data Ratio		1:1:1:1:1					1:1:0.5:1:0.5					1:1:1:1:0.1				
	Domain	Scheme	0%	25%	50%	75%	100%	0%	25%	50%	75%	100%	0%	25%	50%	75%	100%
AD	target (A)	DADA	94.4%	96.7%	96.8%	96.9%	97.1%	93.9%	96.0%	96.1%	96.2%	96.4%	89.5%	91.2%	91.7%	92.8%	93.1%
		ADA	93.5%	94.9%	95.5%	95.4%	95.2%	88.1%	89.7%	91.7%	93.3%	94.2%	87.2%	89.7%	89.2%	88.5%	88.3%
		MK-MMD	91.9%	92.3%	93.8%	95.9%	96.9%	91.1%	91.7%	93.8%	95.1%	96.3%	87.0%	88.9%	90.5%	91.8%	93.1%
		MMD	83.2%	83.9%	85.1%	89.3%	97.0%	83.7%	85.1%	87.7%	91.3%	96.2%	84.9%	85.7%	88.2%	90.7%	93.0%
	target (B)	DADA	95.1%	97.4%	97.6%	97.8%	98.1%	94.7%	96.8%	97.6%	97.8%	97.9%	92.1%	93.7%	94.0%	94.7%	95.4%
		ADA	94.9%	95.6%	96.3%	95.9%	95.5%	89.3%	90.9%	93.3%	95.1%	96.1%	89.3%	90.5%	93.7%	93.4%	92.9%
		MK-MMD	93.3%	93.9%	94.4%	96.1%	97.5%	92.8%	93.1%	94.4%	96.1%	97.5%	89.4%	91.5%	92.1%	93.6%	95.4%
		MMD	87.6%	88.7%	91.0%	94.8%	97.4%	86.6%	87.6%	91.0%	94.8%	97.1%	85.1%	86.6%	89.1%	92.2%	95.4%

**Table 7 sensors-21-08475-t007:** Time cost of all schemes.

Schemes	Processing Time (s)	Time Cost per Epoch (s)		
		Pre-Train	Fine-Tuning	Total
AD-DADA	97	162	102	264
AD-ADA			97	259
AD-MK-MMD			147	309
AD-MMD			111	273
DANGR	41	101	84	185
WiLlSensing	59	84	69	153

## Data Availability

Data sharing is not applicable to this article.

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
