# Peer review of "A Semi-Supervised Transfer Learning with Dynamic Associate Domain Adaptation for Human Activity Recognition Using WiFi Signals†"

_sensors, 2021, doi:10.3390/s21248475_

Round 1

Reviewer 1 Report

Automatic recognition of human activity is an important field in providing personalized care mainly in medicine and sociology through human-computer interaction analysis. Although there are many developments in this area, there is still no solution that has a cost-performance ratio that allows easy implementation in a real building environment. This study is on a topic of relevance and general interest to the readers of the journal.

This paper addresses the problem of recognizing human activity independent of the environment, also known as domain adaptation. The authors use the channel state information (CSI) of WiFi signals. They propose semi-supervised transfer learning with dynamic adaptation of the associate domain in order to recognize human activity. To improve the recognition accuracy at the data preprocessing stage, missing packet filling, noise removal, background estimation, feature extraction and feature enhancement are performed. The pre-trained model is trained from the source domain by collecting a complete set of labeled data for all human activity patterns of the CSI. The pre-trained model is then transferred to the target environment through the semi-supervised transfer learning stage. The authors propose an algorithm for dynamically adapting the associated domain called DADA. The advantage of DADA is that it provides a dynamic strategy to remove different effects in different environments. An attention-based DenseNet (AD) model is developed as a training network, which is modified from the existing DenseNet by adding the attention feature. The proposed solution (DADA-AD) has been tested for five types of human activities (falling, standing, squatting, jumping and sitting). The recognition accuracy of these activities in the test environment chosen by the authors is 97.4%.

The article as a whole is very well written. The content by sections is balanced. I cannot recommend this paper for publication as it stands, as I have comments on the experimental section. I therefore recommend that a revision be made. I explain my comments in more detail below.

Major comments:

1. Important information about the experiments conducted is missing. What is the number of participants in the experiments? What is their height and weight. Does the recognition accuracy of human activity depend on these parameters?

2. There is no analysis of the number of WiFi transmitters and receivers selected and their position in the building space. It is also not clear how the human activity recognition accuracy changes as the distance between the transmitter and receivers increases.

3. The authors would do well to analyse what disadvantages the proposed solution has. It should be analysed what exactly the proposed solution can be used for and what it is not applicable for. It should also be analysed whether it is possible to implement this solution in a real building environment and under what conditions.

4. The sentence "The sampling rate is 1kHz and sampling is 3 seconds, the CSI matrix size is 90 × 3000" (472-473) should be rewritten to make it more understandable to readers.

5. Labels a), b) and C) are missing for Fig. 16. Only source and target domains should remain on this figure. The picture of the Intel 5300 NIC controller is redundant. The location of the WiFi transmitter and receivers should be on another figure. 

Author Response

Automatic recognition of human activity is an important field in providing personalized care mainly in medicine and sociology through human-computer interaction analysis. Although there are many developments in this area, there is still no solution that has a cost-performance ratio that allows easy implementation in a real building environment. This study is on a topic of relevance and general interest to the readers of the journal.

This paper addresses the problem of recognizing human activity independent of the environment, also known as domain adaptation. The authors use the channel state information (CSI) of WiFi signals. They propose semi-supervised transfer learning with dynamic adaptation of the associate domain in order to recognize human activity. To improve the recognition accuracy at the data preprocessing stage, missing packet filling, noise removal, background estimation, feature extraction and feature enhancement are performed. The pre-trained model is trained from the source domain by collecting a complete set of labeled data for all human activity patterns of the CSI. The pre-trained model is then transferred to the target environment through the semi-supervised transfer learning stage. The authors propose an algorithm for dynamically adapting the associated domain called DADA. The advantage of DADA is that it provides a dynamic strategy to remove different effects in different environments. An attention-based DenseNet (AD) model is developed as a training network, which is modified from the existing DenseNet by adding the attention feature. The proposed solution (DADA-AD) has been tested for five types of human activities (falling, standing, squatting, jumping and sitting). The recognition accuracy of these activities in the test environment chosen by the authors is 97.4%.

Answer:

Thank you for fully understanding our paper and make a simple report to clearly summarize our contribution.

The article as a whole is very well written. The content by sections is balanced. I cannot recommend this paper for publication as it stands, as I have comments on the experimental section. I therefore recommend that a revision be made. I explain my comments in more detail below.

Major comments:

  1. Important information about the experiments conducted is missing. What is the number of participants in the experiments? What is their height and weight. Does the recognition accuracy of human activity depend on these parameters?

Answer:

Thank you for your insightful comment. We collected 5000 data from the source domain and 4000 data from two target domains, which are obtained by 11 participants in the laboratory, with a height of about 160 to 180 cm and a weight of about 65 kg to 100 kg. The work investigates five kinds of HAR patterns, including squat, fall, stand, sit, and jump. The considered HAT patterns in this paper belong to the coarse-grained human activity pattern but are not fine-grained human activity patterns. Therefore, the recognition accuracy of human activity is not directly depended on the mentioned parameters.

  1. There is no analysis of the number of WiFi transmitters and receivers selected and their position in the building space. It is also not clear how the human activity recognition accuracy changes as the distance between the transmitter and receivers increases.

Answer:

Thank you for your insightful comment. First, Fig. 18(b) had shown that the pre-training accuracy, which is the recognition accuracy to predict correct HAR pattern from five kinds of HAR patterns in the same source environment, while the pre-trained model is trained from a source domain. These results reported that the more pairs of WiFi transmitters and receivers utilized, the higher human activity recognition accuracy is.

Many recent works of recognizing human activity under the environment-dependent. In general, the distance between the transmitter and receivers as increases, the lower human activity recognition accuracy will be. In our work, we mainly investigate the problem of recognizing human activity under the environment-independent. To more easily understand the effects and our contribution of environment-independent HAR, we need to fixed some parameters; the number of WiFi transmitters and receivers and distance between the transmitter and receivers when moving to the new target domains, and only focus on the parameters under consideration of the different ratios of labeled/unlabeled datasets and under the consideration of the data imbalance problem. Thanks for the reviewers’ useful comments, we will add these assumptions in our experimental setup to make the readers more easily understand.

  1. The authors would do well to analyse what disadvantages the proposed solution has. It should be analysed what exactly the proposed solution can be used for and what it is not applicable for. It should also be analysed whether it is possible to implement this solution in a real building environment and under what conditions.

Answer:

Thank you for your insightful comment. The disadvantages of WLAN-sensing HAR are discussed in the Conclusion and also describe the possibility to implement this WLAN-sensing solution in a real building environment.

  1. The sentence "The sampling rate is 1kHz and sampling is 3 seconds, the CSI matrix size is 90 × 3000" (472-473) should be rewritten to make it more understandable to readers.

Answer:

Thank you for your insightful comment. We have rewritten this sentence to make it more understandable to readers.

  1. Labels a), b) and C) are missing for Fig. 16. Only source and target domains should remain on this figure. The picture of the Intel 5300 NIC controller is redundant. The location of the WiFi transmitter and receivers should be on another figure. 

Answer:

Thank you for your insightful comment. We have added the labels a), b) and C) in Fig. 16, and only source and target domains remain on Fig. 16. The picture of the Intel 5300 NIC controller is removed due to its redundancy. The location of the WiFi transmitter and receivers is solely moved to a new Fig. 17 in the revised paper.

Reviewer 2 Report

The overall idea of the manuscript is clear, the structure is rigorous, the content is full and the description is detailed. The experiments are adequate, the data are real and credible, and the results show that the methods in the manuscript are feasible, can solve the practical problems, and have certain innovation.However, the following deficiencies exist in the manuscript.

  1. In the introduction section, both the first and second major contributions of this manuscript discuss the problems solved by DADA and its advantages. It is suggested that these two paragraphs should be revised to make the main idea clear and well-layered.
  2. The conclusion section is too brief and does not completely include all the outstanding contributions of the manuscript.In addition, the topic of the article is the implementation of environment-independent human activity recognition, while the conclusion is silent on the activity recognition results of the proposed method. Please write the summary carefully and make sure to highlight the novelty of the article, the advantages of the proposed method and the experimental validation results.
  3. Some of the contents in the introduction are not logically clear and the lines are not smooth, such as lines 49-55, 59-61, 117-119. It is suggested that the content of the manuscript be carefully checked to ensure that the lines are smooth and logically clear.
  4. 177 and 178 lines are confusingly expressed, so please check them carefully.
  5. "sections 3.1, 3.2, and 3.3" in line 281 and "5.1" and "5.2" in line 658 do not match the actual section numbers. There are many inconsistencies between the chapter numbers and the references in the content of the manuscript, so please check the structure of the article carefully.
  6. Should Fig. 5(a) in line 395 and Fig. 5(b) in line 400 be Fig. 6(a) and Fig. 6(b)?
  7. In line 442, the author says that the definition of CSI vectors is in section 3.2, while it is actually in section 2.2.So please double check the manuscript and make sure the context is consistent.

Author Response

The overall idea of the manuscript is clear, the structure is rigorous, the content is full and the description is detailed. The experiments are adequate, the data are real and credible, and the results show that the methods in the manuscript are feasible, can solve the practical problems, and have certain innovation. However, the following deficiencies exist in the manuscript.

Answer:

Thank you for your insightful comment and fully understanding the contribution of our paper.

  1. In the introduction section, both the first and second major contributions of this manuscript discuss the problems solved by DADA and its advantages. It is suggested that these two paragraphs should be revised to make the main idea clear and well-layered.

Answer:

Thank you for your insightful comment. We have re-written the first and second major contributions of this manuscript discuss the problems solved by DADA and its advantages to make the main idea clear and well-layered.

  1. The conclusion section is too brief and does not completely include all the outstanding contributions of the manuscript. In addition, the topic of the article is the implementation of environment-independent human activity recognition, while the conclusion is silent on the activity recognition results of the proposed method. Please write the summary carefully and make sure to highlight the novelty of the article, the advantages of the proposed method, and the experimental validation results.

Answer:

Thank you for your insightful comment. We have rewritten the conclusion to completely include all the outstanding contributions of our manuscript, and describe the implementation of environment-independent human activity recognition, and highlight the novelty of this work, the also discuss the advantages of the proposed method and the experimental validation results.

  1. Some of the contents in the introduction are not logically clear and the lines are not smooth, such as lines 49-55, 59-61, 117-119. It is suggested that the content of the manuscript be carefully checked to ensure that the lines are smooth and logically clear.

Answer:

Thank you for your insightful comment. We had re-written and carefully checked some of the contents in the introduction to make the contents more clear, especially for lines 49-55, 59-61, 117-119 in the original version, to ensure that the lines are smooth and logically clear.

  1. 177 and 178 lines are confusingly expressed, so please check them carefully.

Answer:

Thank you for carefully checking the problem in 177 and 178 lines by misusing \section and \subsection. We have reorganized our paper organization to resolve the confusion of expression in lines 177 and 178 lines.

  1. "sections 3.1, 3.2, and 3.3" in line 281 and "5.1" and "5.2" in line 658 do not match the actual section numbers. There are many inconsistencies between the chapter numbers and the references in the content of the manuscript, so please check the structure of the article carefully.

Answer:

Thank you for carefully checking the wrong section number problem, this revised paper fixes all inconsistencies, including "sections 3.1, 3.2, and 3.3" in line 281 and "5.1" and "5.2" in line 658 to match the actual section numbers.

  1. Should Fig. 5(a) in line 395 and Fig. 5(b) in line 400 be Fig. 6(a) and Fig. 6(b)?

Answer:

Thank you for carefully checking the figure number problem. Yes, we have corrected Fig. 5(a) in line 395 (original version) and Fig. 5(b) in line 400 (original version) to be Fig. 6(a) and Fig. 6(b).

  1. In line 442, the author says that the definition of CSI vectors is in section 3.2, while it is actually in section 2.2.So please double check the manuscript and make sure the context is consistent.

Answer:

Thank you for carefully checking the inconsistent problem, since we corrected the wrong section number in the revised paper, “the definition of CSI vectors is in section 3.2” is still reserved to keep the content consistent.
